# SARS-CoV-2 impairs male fertility by targeting semen quality and testosterone level: A systematic review and meta-analysis

**Ashonibare V. J.[1,2], Ashonibare P. J.[2,3], Akhigbe T. M.[2,4], R. E. Akhigbe [2,3]***

**1** Medical Faculty, Department of Cardiovascular Surgery and Research Group for Experimental Surgery, Cardiovascular Regenerative Medicine and Tissue Engineering 3D Lab, Heinrich Heine University, Düsseldorf, Germany, **2** Reproductive Biology and Toxicology Research Laboratory, Oasis of Grace Hospital, Osogbo, Nigeria, **3** Department of Physiology, Ladoke Akintola University of Technology, Ogbomosho, Oyo State, Nigeria, **4** Department of Agronomy, Breeding and Genetic Unit, Osun State University, Osun State, Nigeria

* akhigberoland@gmail.com, reakhigbe@lautech.edu.ng

**Data Availability Statement:** All data are in the paper and/or Supporting Information files.

**Funding:** The author(s) received no specific funding for this work.

## Abstract

### Background

Since the discovery of COVID-19 in December 2019, the novel virus has spread globally causing significant medical and socio-economic burden. Although the pandemic has been curtailed, the virus and its attendant complication live on. A major global concern is its adverse impact on male fertility.

### Aim

This study was aimed to give an up to date and robust data regarding the effect of COVID-19 on semen variables and male reproductive hormones.

### Materials and methods

Literature search was performed according to the recommendations of PRISMA. Out of the 852 studies collected, only 40 were eligible for inclusion in assessing the effect SARS-CoV-2 exerts on semen quality and androgens. More so, a SWOT analysis was conducted.

### Results

The present study demonstrated that SARS-CoV-2 significantly reduced ejaculate volume, sperm count, concentration, viability, normal morphology, and total and progressive motility. Furthermore, SARS-CoV-2 led to a reduction in circulating testosterone level, but a rise in oestrogen, prolactin, and luteinizing hormone levels. These findings were associated with a decline in testosterone/luteinizing hormone ratio.

**Competing interests:** The authors have declared that no competing interests exist.

## Conclusions

The current study provides compelling evidence that SARS-CoV-2 may lower male fertility by reducing semen quality through a hormone-dependent mechanism; reduction in testosterone level and increase in oestrogen and prolactin levels.

## Introduction

Severe acute respiratory syndrome coronavirus 2 (SARS-CoV-2), which is implicated as the causative organism of the Corona-Virus disease 2019 (COVID-19) has remained a global concern since its outbreak [1–3]. SARS-CoV-2 is a sheathed β-coronavirus, which is genetically similar to SARS-CoV-1 (80%) and 96.2% with Bat coronavirus RaTG13 [4]. The S protein contains the S1 sub-unit, which carries the receptor binding domain that tethers to the angiotensin-converting enzyme 2 (ACE 2) [5,6], and facilitates binding to and entry into host cells [4,6]. Though quite similar, SARS-CoV-2 spreads more expeditiously than SARS-CoV-1, as it has a higher net reproductive rate. Additionally, SARS-CoV-2 exhibits stronger binding to its host receptor cells and greater host invasion because of its slight structural difference from SARS-CoV-1 [7,8]. However, angiotensin-converting enzyme 2 (ACE2) is the primary host receptor of SARS-CoV [4]. It is liberally present in the epithelial tissue of the lung and small intestine, heart, lungs, kidneys, and testes in humans [9–19], and may contribute possible entry portal for SARS-CoV [20].

As of May 2023, over 766 million COVID- 19 cases, with about 7 million mortalities were reported [9]. Studies have revealed that COVID-19 mainly affects both male and female respiratory systems [4,8]. Studies have also demonstrated that the virus causes damage to multiple organs, including the kidney, heart, liver, brain [10,12], and testes [2,4,6,8,13]. In addition, there is proof that SARS-CoV-1 exerts a more severe impact on males than females [6,14–17]. Also, orchitis has been reported in males recovering from the SARS virus [3,18]. Despite this, findings on the adverse effect of this deadly virus on the male reproductive system are limited and contentious. In a systematic review and meta-analysis by Corona et al. [21], SARS-CoV-2 infection was linked with low semen quality and serum testosterone level. This is in agreement with earlier systematic review and meta-analysis by Tiwari et al. [22]. The study however had some frailties- first, the random-effect model was used irrespective of the level of diversity, which might affect the findings of the meta-analysis. Also, no sensitivity analyses were performed to rule out the influence of diversity. Finally, the authors failed to apply the finding of the quality of the appraised studies to their analysis.

Therefore, the aim of this study is oriented towards providing an overhauling meta-analysis on the consequence of COVID-19 on male fertility. This review gives an insight into how COVID-19 impact semen quality and male reproductive hormones to modulate male fertility. So far as we are aware, this research pioneers the evaluation of the impact of COVID-19 by comparing between infected and non-infected subjects, before and after treatment in infected patients, and infected and pre-COVID state in the same patients. Hence, the present study evinces a robust review and analysis of the influence of SARS-CoV-2 on male fertility.

## Materials and methods

### Protocol and eligibility criteria for inclusion

This study was registered on Prospero (CRD42024533906). This study was conducted on published works that evaluated the influence of SARS-CoV-2 on male fertility. The study adopted

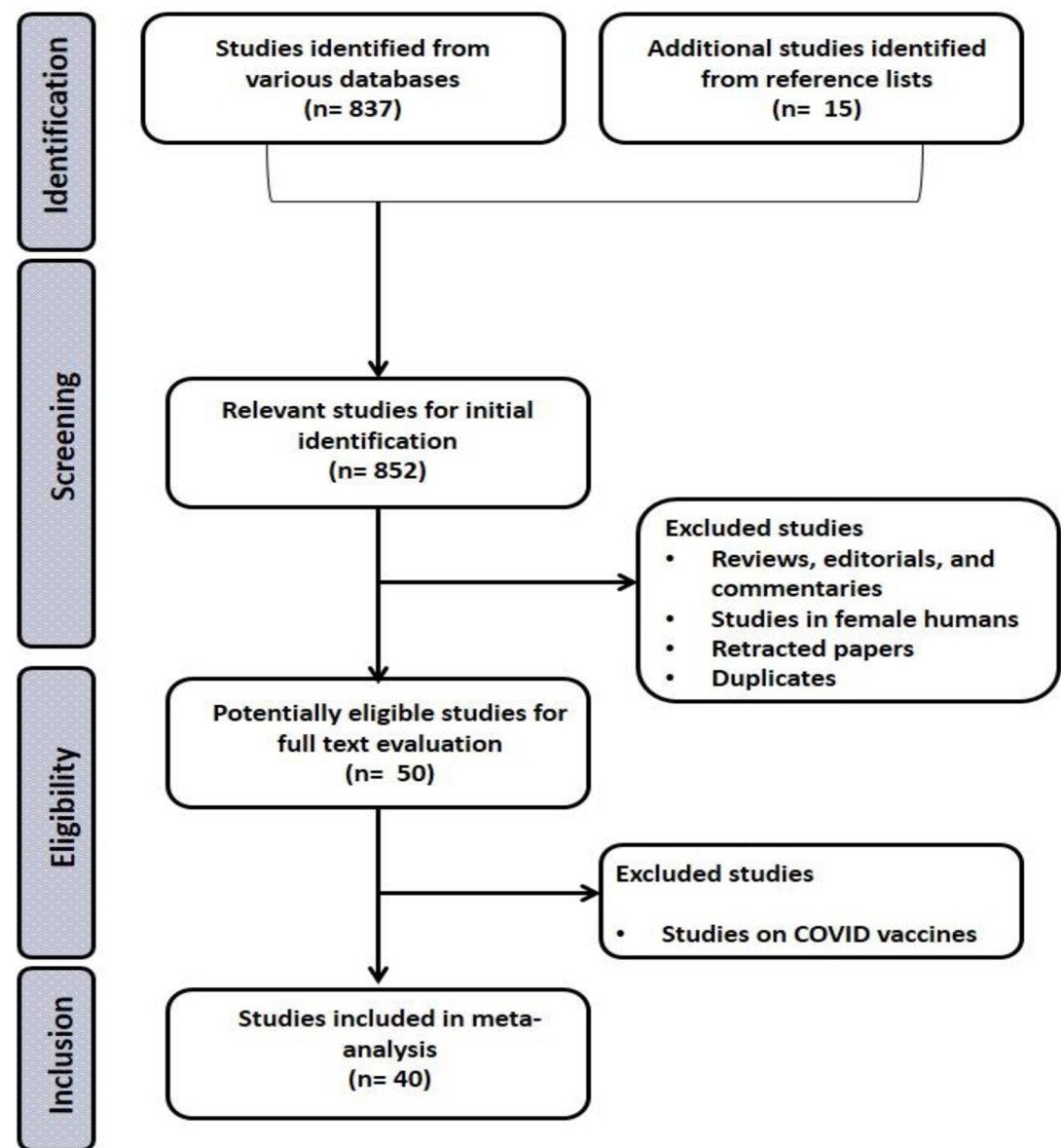

**Fig 1. PRISMA flowchart for the strategic identification, screening, and inclusion of eligible studies.**

the "Preferred Reporting Items for Systematic Reviews and Meta-analyses (PRISMA)" strategy, which is provided as Fig 1.

This study adopted the Population, Exposure, Comparator/Comparison, and Outcomes (PECO) model. All studies published until October 2023 that were eligible based on set criteria, were collected. The studied populations were male in their reproductive ages, who had an exposure to SARS-CoV-2 and developed COVID 19. The studies were either retrospective or prospective among COVID 19-infected patients with age-matched control who were COVID 19-negative. In cases where there were no COVID 19-negative control groups, outcomes before and after the treatment of COVID 19 or at pre-COVID and COVID 19-infected states should be presented. The outcome measured were conventional semen parameters viz. ejaculate volume, sperm count, concentration, viability, normal morphology, total and progressive

motility, and seminal fluid leukocyte level, and male reproductive hormones namely testosterone (T), oestrogen, prolactin, follicle-stimulating hormone (FSH), and luteinizing hormone (LH) levels. T/LH and FSH/LH were also measured.

Exclusion criteria included absence of a comparator as control, studies in females, *in vitro* studies, commentaries, review articles, letters to editor, editorials, preprint, conference abstracts, retracted papers, and degree thesis. No language or country restriction was applied.

## Search strategy

An organized search using EMBASE, Pubmed/MEDLINE, Scopus, and Web of Science databases was performed. The keywords combined were "COVID", "COVID 19", "coronavirus", "SARS-CoV-2", "semen", "semen analysis", "seminal fluid", "sperm", "sperm cells", "spermatozoa", "sperm parameter", "sperm variable", "sperm count", "sperm concentration", "sperm viability", "sperm vitality", "sperm motility", "total sperm motility", "progressive sperm motility", "sperm morphology", "semen volume", "ejaculate volume", "seminal leukocyte", and "seminal WBC', "luteinizing hormone", "LH", "follicle stimulating hormone", "FSH", "testosterone", "male fertility", "male infertility", "male reproduction". Abstracts and full text of articles collected were independently evaluated for eligibility by AVJ, APJ, and. ATM, and differences of opinion were resolved by ARE.

## Data collection, assessment of quality of eligible studies, and meta-analysis

The eligible studies were appraised for quality and data collected by AVJ, APJ, and. ATM. Disputes were resolved by ARE. Data gathered from the appropriate studies include the last name of the principal investigator, publication date, country of study origin, study design, method of COVID 19 diagnosis, sample size and ages of patients, duration of infection, and measured outcomes of interest. The outcomes of interest were pull out as mean and standard deviation. When the variables were presented in other forms, the mean and standard deviation were derived from the provided data. In cases where the outcomes were reported in Figs, they were converted to values using Web Plot Digitizer.

The quality of evidence in the eligible papers was evaluated using the ErasmusAGE quality score for systematic reviews, which assigns a number between 0 and 2 to five domains [23]. Furthermore, the "Office of Health Assessment and Translation (OHAT)" methodology was used to evaluate the risk of bias (RoB) [24]. Using the "Grading of Recommendations Assessment, Development and Evaluation (GRADE) Working Group" standards as a guide, the "OHAT approach for systematic review and evidence integration for literature-based health assessment was used to assess the certainty of the evidence" [25,26].

Review Manager (version 5.4.1) was used to conduct the quantitative meta-analyses. From the eligible studies, the standardized mean difference (SMD) at 95% confidence intervals (CIs) was calculated. A random-effect model was used when P-value < 0.1 or I2 > 50% which indicates the existence of significant variety; otherwise, a fixed-effect model was utilized. To assess the possible sources of diversity, sensitivity analysis was conducted by excluding the studies with the largest weight, high RoB (< 4), low quality of evidence (< 5) and low certainty of evidence. Also, the generated funnel's plots were visually assessed for publication bias.

## Results

### The selection of studies and the attributes of the relevant studies

Out of the 852 publications screened, only 50 were potentially eligible for evaluation. Finally, 40 studies [27–66] were deemed eligible for inclusion in this study (Fig 1). The eligible papers

were published between 2020 and 2023, and they were from China (7), Germany (1), India (1), Indonesia (1), Iran (6), Iraq (2), Italy (5), Jordan (2), Russia (1), Turkey (12), UK (1), and USA (1). The data collected included the surname of the year of publication, principal investigator, country of study origin, study design, method of diagnosing COVID-19, studied population size, participants'/patients' age range, duration of infection, outcomes measured (Table 1).

## Assessment of the quality of evidence, RoB, and certainty of evidence

A larger part of the studies had good quality of evidence, except 7 of them [27,31,40,48,50,56,64] that had low quality of evidence (<5) (Table 2). Also, the included studies had moderate (4/9-6/9) to low (>6/9) RoB (Table 3). In addition, the certainty of evidence in the included studies were moderate to high, except in 3 studies [29,48,56] with low certainty of evidence (Table 4).

## Meta-analysis and sensitivity analysis

**Ejaculate volume.** Based on the details of the meta-analysis of the 13 eligible studies that compared ejaculate volume in 591 COVID-positive patients with 722 COVID-negative individuals, SARS-CoV significantly reduced the ejaculate volume of infected patients (SMD -0.38 [95% CI: -0.70, -0.05] $P = 0.02$). Also, a marked inter-study diversity was noted ($I^2 = 85\%$; $X^2$ $P < 0.00001$). Sensitivity analysis showed that ejaculate volume was still significantly reduced in SARS-CoV-infected patients when compared with the SARS-CoV-negative ones (SMD -0.42 [95% CI: -0.77, -0.07] $P = 0.02$), and the inter-study diversity was also significant ($I^2 = 85\%$; $X^2$ $P < 0.00001$) (Fig 2A). Furthermore, the comparison of 286 COVID-positive patients before treatment with 300 patients after treatment revealed that the ejaculate volume was significantly increased after treatment when compared to before treatment (SMD -0.30 [95% CI: -0.46, -0.14] $P = 0.0003$), and there was no significant inter-study diversity ($I^2 = 36\%$; $X^2$ $P = 0.13$). However, sensitivity analysis demonstrated that the ejaculate volume was not different before and after COVID treatment (SMD -0.24 [95% CI: -0.59, 0.11] $P = 0.19$). This showed marginal significant inter-study diversity ($I^2 = 55\%$; $X^2$ $P = 0.05$) (Fig 2B). More so, it was observed that SAR-Cov-2 infection significantly reduced ejaculate volume of patients when compared with their pre-COVID (SMD -0.28 [95% CI: -0.55, -0.01] $P = 0.04$). There was a significant inter-study diversity ($I^2 = 67\%$; $X^2$ $P = 0.004$). This significant difference persisted even after a sensitivity analysis (SMD -0.29 [95% CI: -0.55, -0.03] $P = 0.03$), and there was no significant inter-study diversity ($I^2 = 35\%$; $X^2$ $P = 0.20$) (Fig 2C). The publication bias is shown in Fig 3.

**Sperm count.** SARS-CoV-2 infection significantly reduced sperm count in contrast to non-infected persons (SMD -0.74 [95% CI: -1.43, -0.06] $P = 0.03$), and there was a marked heterogeneity between studies ($I^2 = 95\%$; $X^2$ $P < 0.00001$); however after sensitivity analysis, SARS-CoV-2 infection only led to a marginal decline in sperm count (SMD -0.90 [95% CI: -1.91, 0.10] $P = 0.08$), and we observed a marked heterogeneity between studies ($I^2 = 96\%$; $X^2$ $P < 0.00001$) (Fig 4A). However, COVID-19 treatment did not significantly improve sperm count when compared with the pre-treatment value (SMD -0.24 [95% CI: -0.66, 0.17] $P = 0.24$), and there was a marked heterogeneity between studies ($I^2 = 83\%$; $X^2$ $P < 0.00001$), which persisted after sensitivity analysis (SMD -0.20 [95% CI: -0.78, 0.38] $P = 0.50$) with no marked heterogeneity between studies ($I^2 = 83\%$; $X^2$ $P < 0.00001$) (Fig 4B). Nonetheless, SARS-CoV-2 infection significantly reduced sperm count when compared with the pre-COVID value of the patients (SMD -0.27 [95% CI: -0.45, -0.10] $P = 0.002$), and there no substantial inter-study diverseness was found ($I^2 = 37\%$; $X^2$ $P = 0.16$) (Fig 4C). The funnels' plots showing the publication bias are presented in Fig 5.

**Table 1. Eligible studies included in the meta-analysis that reported the effects of COVID-19 on semen quality and male sex hormones.**

| References | Study design | Country | Diagnosis of COVID-19 | Examined population | Age (years) | Duration of infection (months) | Outcomes/variables measured | |
|---|---|---|---|---|---|---|---|---|
| | | | | | | | Semen | Hormone |
| Abbas et al., 2022 [27] | Cross-sectional | Baghdad/Iraq | - | COVID-19 (70) Control (50) | 25–55 | - | - | LH, FSH, Prolactin↑ |
| Aksak et al., 2022 [28] | Cross-sectional | Adan/Turkey | PCR | COVID-19 (100) Control (100) | 20–50 | 4–12 | Semen volume, concentration, motility, morphology | - |
| Al-Alami et al, 2022 [29] | Retrospective | Jordan | - | Vaccinated (28) Vaccinated and infected (14) Neither vaccinated nor infected (3) Infected only (4) N = 49 N´ = 354 | - | - | sperm concentration, sperm progressive motility,semen liquefaction time, ejaculate volume, normal forms existing within the semen, and ejaculate viscosity. | - |
| Al-Bashiti et al, 2022 [30] | Cross sectional | Amman, Jordan | PCR | COVID(81) Control (76) | 54.35±14.46 (COVID) 49.59±15.80 (Control) 20–80 | - | - | Testosterone↓ inhibin B ↓ |
| Azzawi and Abdulrahman, 2022 [31] | Cross sectional | Fallujah, Iraq | - | Recovered (60) Control (30) | 20–49 | - | - | PSA, testosterone ↓, FSH↑, LH ↑ |
| Best et al, 2021 [32] | Prospective | Florida, USA | PCR | COVID (30) Control (30) | 40 (IQR = 24.75) (COVID) 42 (IQR = 9.8) (Control) 18–70 | 90 days follow up | Volume, pH, concentration ↓, total sperm number ↓ | - |
| Camici et al., 2021 [33] | Retrospective cross-sectional | Rome, Italy | PCR | COVID (24) Control (24) | 18- 65YRS Control: (43–57) COVID: (43–59) | 2months - | | androstenedione, 5α-dihydrotestosterone, Oestradiol, sex hormone binding globulin, testosterone |
| Cinislioglu et al, 2022 [34] | Prospective | Erzurum, Turkey | PCR | COVID (358) Control (92) | 64.9 (11.6) (COVID) 67.2 (13.6) (Control)25–91 | 7months | - | Testosterone ↓ FSH ↑ LH ↑ TT:LH ↓ |
| Dipankar et al, 2022 [35] | Prospective/ Longitudinal | Patna, India | PCR | 30 | 19–45 | 74 days follow up | Volume↑, viscosity↓, agglutination ↓, liquefaction time ↓, pH, volume, progressive motility ↑, total motility ↑ sperm count ↑, total sperm count ↑, morphology↑, tail defect, head defect↓, neck defect, DNA Fragmentation Index (DFI) ↓, cytoplasmic droplet ↑, vitality↑, fructose present↑, normal morphology, WBC ↓ | - |

*(Continued)*

**Table 1.** (Continued)

| References | Study design | Country | Diagnosis of COVID-19 | Examined population | Age (years) | Duration of infection (months) | Outcomes/variables measured | |
|---|---|---|---|---|---|---|---|---|
| | | | | | | | Semen | Hormone |
| Enikeev et al, 2022 [36] | Prospective | Moscow, Russia | PCR | COVID on admission (44), COVID at 3 months of follow up (37), Control (44) | 46.7±9.9 (COVID) 30.7±9.8 (Control) 18–65 | 3 months follow up | Concentration ↑, total sperm count volume, (total motility, progressive motility, slow progressive motility, non-progressive motility)↑ rapid progressive motility, no motility ↓immobile sperm ↑ vitality ↓ normal morphology ↓wbc ↑ Agglutination, pH, normal morphology | IIEF-5, Prolactin, FSH, LH ↓, Testosterone ↑ |
| Erbay et al, 2021 [37] | Retrospective, CS | Instabul, Turkey | PCR | COVID-(19) 69 | 20–45 | 74 days | Volume, concentration, vitality, sperm number, total motility, progressive motility | - |
| Falahieh et al, 2021 [38] | | Urmia, Iran | PCR | 20 | 20 and 50 | 14, 120 days | volume, colour, viscosity and pH of the semen sample, sperm concentration, total, progressive motility↑, normal morphology ↑ and viability | - |
| Gacci et al, 2021 [39] | Prospective cross-sectional | - | PCR | 43 Nonhospitalzed (mild) Hospitalized (moderate) ICU (severe) | 18–65 | - | Volume, cell number ↑, concentration, progressive motility, vitality ↓, normal morphology, pH, | - |
| Gul et al, 2021 [40] | Cross sectional | Bursa, Turkey | SARS-CoV-2 nucleic acid test | 29 | 18–41 | ? | Semen volume, sperm concentration, total sperm count, total motility, progressive motility | Testosterone, FSH, LH, prolactin |
| Guo et al, 2021 [41] | Prospective | Anhui, China | PCR | COVID-19 (41) Control (50) | COVID-19: 26.0 (22.0–34.0) Control: 26.5 (25.0–34.0) | ? | Concentration, volume, total sperm count, abnormal morphology, vitality, sperm motility, progressive motility, motile sperm count | Estradiol, FSH, LH, progesterone, testosterone (T), prolactin, anti Mullerian hormone (AMH) and inhibin B |
| Hadisi et al, 2022 [42] | Cross sectional | Ahar, Iran | PCR | COVID-19 (60) Control (60) | ? | - | - | estradiol, FSH, LH, prolactin, progesterone, testosterone, cortisol and thyroid stimulating hormone (TSH) |

(Continued)

**Table 1.** (Continued)

| References | Study design | Country | Diagnosis of COVID-19 | Examined population | Age (years) | Duration of infection (months) | Outcomes/variables measured | |
|---|---|---|---|---|---|---|---|---|
| | | | | | | | Semen | Hormone |
| Hamarat et al, 2022 [43] | Prospective, longitudinal | Konya, Turkey | PCR | 41 | 22–46 | Over 70 days | sperm concentration ↓, total sperm number ↓, semen volume ↓, sperm motility (progressive motility, non-progressive motility, and immotility percentages),normal morphology ↓, head↑, neck, and tail anomaly ↓ | - |
| Holtmann et al, 2020 [44] | Cross sectional | Duesseldorf, Germany | PCR | Control: 14 Mild case: 14 Moderate: 2 | Control: 33.4 ±13.1 Mild case: 42.7 ±10.4 Moderate: 40.8±8.7 | - | Volume, concentration, total sperm number, sperm number, progressive motility, complete motility, immotility, | - |
| Hu et al, 2022 [45] | Prospective | Wuhan, China | PCR | COVID (36) Control (45) | 31.75±5.77 31.49±3.10 (NS) | - | PH, volume, sperm concentration, total sperm number, progressive motility and total motility | - |
| Kadihasanoglu et al, 2021 [46] | Prospective cross sectional | Istanbul, Turkey | PCR | COVID-19 (89), controls (143). | COVID: 49.9 ± 12.5 Control: 50 ± 7.8 20 and 65 | - | - | Testosterone, LH, FSH, and prolactin. |
| Karkin & Gürlen, 2022 [47] | Cross sectional | Adana, Turkey | PCR | 348 | 20–74 | - | | TT, LH, FSH |
| Koç & Keseroğlu, 2021 [48] | Prospective cross sectional | Ankara, Turkey | PCR | COVID (21) | 32±6.30 | 5Days | semen volume, percentage of total motility, percentage of progressive motility, and normal sperm morphology | TT, LH, FSH |
| Kumar et al., 2023 [49] | Cross sectional | Patna, India | - | Pre COVID (102 COVID (137). | 33.1 (6.7) | - | sperm concentration, total sperm count, percentage of total motility, percentage of cells with residual cytoplasm, and the percentages of head and tail defects | |
| Li et al, 2020 [50] | Cross sectional | Wuhan, China | PCR | Control (22) COVID(23) | 27–55 | Control: 40.5§5.9 COVID: 40.8§8.5 | Sperm concentration | - |

(*Continued*)

**Table 1.** (Continued)

| References | Study design | Country | Diagnosis of COVID-19 | Examined population | Age (years) | Duration of infection (months) | Outcomes/variables measured Semen | Hormone |
|---|---|---|---|---|---|---|---|---|
| Livingstone et al, 2022 [51] | Cross sectional | Walsall (United Kingdom) | PCR | Control (25) COVID (85) | Control: 68 (56–85) COVID: 75 (64–85) | - | - | Testosterone |
| Ma et al, 2021 [52] | Prospective cross sectional | Zhongnan Hubei Province, China. | PCR | Control (273) COVID (119) | Control: 39 (35.0–42.0) COVID: 39 (35.0–44.0) | 3 months | Volume, concentration, vitality, mobile sperm count, non-progressive motility, progressive motility, immotility, normal sperm morphology | Testosterone, oestrogen, FSH, LH, T/LH, T/E2 and FSH/LH |
| Maleki and Tartibian, 2021 [53] | Prospective longitudinal | Tehran, Iran | PCR | Control (84) COVID(105) | 20–40 | 13.2 ± 4.9 days. Till first sampling | semen volume, progressive motility, sperm morphology, sperm concentration, and the number of spermatozoa | - |
| Okçelik 2020 [54] | Prospective | Hacı Bektaş, Turkey | PCR | Control (20) COVID (24) | 18–50 (35.5 ± 9.85) years | 4Months | | FSH, LH and testosterone |
| Paoli et al, 2023 [55] | Retrospective cross sectional | Sapienza, Rome | Nasopharyngeal swab positive for SARS-CoV-2 | COVID-19 (80) Control 1 (98) Control 2 (98) | 18 to 65 (43.9±11.7) | - | Volume, total sperm number, progressive motility, and morphology | FSH, LH, Testosterone |
| Pazir et al, 2021 [56] | Cross sectional | Istanbul, Turkey | PCR | 24 | 18–49 Control: 36.4 ± 13 COVID: 38.2 ± 9.9 | - | Volume, concentration, progressive motility, total motility, mobile sperm count | - |
| Piroozmanesh et al, 2021 [57] | Cross sectional | Qom, Iran | PCR | COVID-19 (60) Control (40) | 20–45 | - | sperm concentration, sperm total motility, sperm vitality, sperm normal forms, and TAC | - |
| Rafiee & Tabei, 2021 [58] | Interventional | Shiraz, Iran | PCR | COVID-19 (100) Control (100) | - | - | sperm concentration, sperm motility, and normal sperm morphology, volume | - |
| Ruan et al, 2021 [59] | Cross sectional | Wuhan, China | PCR | COVID-19 (55) Control (145) | 20–50 Control: 30.69 ±4.36 COVID: 31.15 ±5.32 | - | Semen volumes, sperm concentrations, total sperm counts, motile spermatozoa, morphologically normal spermatozoa, DNA fragmentation index (DFI), | - |

(*Continued*)

**Table 1.** (Continued)

| References | Study design | Country | Diagnosis of COVID-19 | Examined population | Age (years) | Duration of infection (months) | Outcomes/variables measured | |
|---|---|---|---|---|---|---|---|---|
| | | | | | | | Semen | Hormone |
| Salonia et al., 2021 [60] | Cross sectional | Milan, Italy | PCR | Control: 281 COVID: 286 | Control: 46 (35–52) COVID 19: 58 (49–66) | - | - | follicle- stimulating hormone (FSH), luteinizing hormone (LH), tT, and 17β-estradiol (E2) |
| Salonia1 et al., 2022 [61] | Prospective | - | PCR ACE2 | 121 | 49–65 years | 7months | - | Testosterone, oestradiol, LH, FSH |
| Sunnu et al 2022 [62] | Prospective, longitudinal | Surabaya, Indonesia | PCR | 14 | 27–48 | 6 month follow up | semen volume, pH, sperm concentration, total, progressive, non-progressive, and immotile motility percentage | - |
| Temiz et al 2020 [63] | Prospective cross sectional | Istanbul, Turkey | PCR | Control (10) Pre-treatment (10) Post-treatment (10) | 18- to 60 Control: 36.64 ± 9.63 Pre-treatment: 38.00 ± 8.28 Post-treatment: 37.00 ± 8.69 | 4 days | Semen volume, pH, count, concentration, progressive sperm motility, non-progressive sperm motility, total sperm motility, normal morphology | Testosterone, FSH, LH, prolactin, Testosterone/LH, FSH/LH, prolactin/testosterone |
| Vahidi et al 2022 [64] | Cross sectional | Shahid Sadoughi, Iran | PCR | Acute (20) Recovery (20) | 18–45 | - | Sperm count, viability, progressive motility, morphology, immotile, non-progressive | - |
| Wang et al 2022 [65] | Retrospective, | Wuhan, China | PCR | 26 | - | - | Volume, concentration, progressive motility, sperm number, total progressive motility, complete motility, total normal form, normal form, immotile, total number of immotile | FSH |
| Xu et al 2021 [66] | Retrospective cross-sectional | Wuhan, China | SARS-CoV-2 RNA throat swab | COVID-19 (39) Control (22) | Control: 62 (52, 68.75) COVID: 60.0 (46.5, 65.5) | - | - | (testosterone [T], follicle-stimulating hormone [FSH], luteinizing hormone [LH], prolactin [PRL], and estradiol) |

**Sperm concentration.** Analysis of the impact of SARS-CoV-2 on sperm concentration revealed that the novel infection significantly reduced sperm concentration when compared with SARS-CoV-2-uninfected individuals (SMD -0.83 [95% CI: -1.46, -0.20] $P = 0.010$). Again, no substantial heterogeneity between studies was found ($I^2 = 95\%$; $X^2 P < 0.00001$). After sensitivity analysis, SARS-CoV-2 only marginally reduced sperm concentration when compared with individuals who were not SARS-CoV-2 positive (SMD -1.02 [95% CI: -2.16, 0.12] $P = 0.08$). There was a significant inter-study variety ($I^2 = 97\%$; $X^2 P < 0.00001$) (Fig 6A).

**Table 2. Assessment of the quality of evidence of the eligible studies.**

| Study | Study design | Study size | Method of measuring exposure | Method of measuring outcome | Analysis with adjustment | Total |
|---|---|---|---|---|---|---|
| Abbas et al., 2022 [27] | 0 | 1 | 0 | 2 | 0 | 3/10 |
| Aksak et al., 2022 [28] | 0 | 2 | 2 | 2 | 2 | 8/10 |
| Al-Alami et al., 2022 [29] | 0 | 2 | 1 | 1 | 1 | 5/10 |
| Al-Bashiti et al., 2022 [30] | 0 | 2 | 2 | 2 | 0 | 6/10 |
| Azzawi and Abdulrahman, 2022 [31] | 0 | 1 | 0 | 2 | 0 | 3/10 |
| Best et al, 2021 [32] | 1 | 1 | 2 | 2 | 0 | 6/10 |
| Camici et al., 2021 [33] | 0 | 0 | 2 | 2 | 1 | 5/10 |
| Cinislioglu et al., 2022 [34] | 1 | 2 | 2 | 2 | 1 | 8/10 |
| Dipankar et al., 2022 [35] | 1 | 0 | 2 | 2 | 01 | 6/10 |
| Enikeev et al., 2022 [36] | 1 | 1 | 2 | 2 | 01 | 7/10 |
| Erbay et al., 2021 [37] | 0 | 1 | 2 | 02 | 0 | 5/10 |
| Falahieh et al., 2021 [38] | 1 | 0 | 2 | 2 | 0 | 5/10 |
| Gacci et al., 2021 [39] | 0 | 0 | 2 | 2 | 1 | 5/10 |
| Gul et al., 2021 [40] | 0 | 0 | 1 | 1 | 2 | 4/10 |
| Guo et al., 2021 [41] | 1 | 1 | 2 | 2 | 1 | 7/10 |
| Hadisi et al., 2022 [42] | 0 | 1 | 2 | 2 | 1 | 6/10 |
| Hamarat et al., 2022 [43] | 1 | 0 | 2 | 2 | 1 | 6/10 |
| Holtmann et al., 2020 [44] | 0 | 0 | 2 | 2 | 1 | 5/10 |
| Hu et al., 2022 [45] | 1 | 1 | 2 | 2 | 1 | 7/10 |
| Kadihasanoglu et al., 2021 [46] | 1 | 2 | 2 | 2 | 1 | 8/10 |
| Karkin & Gürlen, 2022 [47] | 0 | 2 | 2 | 2 | 1 | 7/10 |
| Koç and Keseroğlu, 2021 [48] | 0 | 0 | 2 | 2 | 0 | 4/10 |
| Kumar et al., 2023 [49] | 1 | 0 | 2 | 2 | 0 | 5/10 |
| Li et al., 2020 [50] | 0 | 0 | 2 | 2 | 0 | 4/10 |
| Livingstone et al., 2022 [51] | 0 | 1 | 2 | 2 | 1 | 6/10 |
| Ma et al., 2021 [52] | 1 | 2 | 2 | 2 | 1 | 8/10 |
| Maleki and Tartibian, 2021 [53] | 1 | 2 | 2 | 2 | 1 | 8/10 |
| Okçelik, 2020 [54] | 1 | 0 | 2 | 2 | 2 | 7/10 |
| Paoli et al., 2023 [55] | 0 | 2 | 2 | 2 | 1 | 7/10 |
| Pazir et al., 2021 [56] | 0 | 0 | 2 | 2 | 0 | 4/10 |
| Piroozmanesh et al., 2021 [57] | 0 | 1 | 2 | 2 | 1 | 6/10 |
| Rafiee and Tabei, 2021 [58] | 2 | 1 | 2 | 2 | 0 | 7/10 |
| Ruan et al., 2021 [59] | 0 | 2 | 2 | 2 | 1 | 7/10 |
| Salonia et al., 2021 [60] | 0 | 2 | 2 | 2 | 0 | 6/10 |
| Salonia1 et al., 2022 [61] | 1 | 2 | 2 | 2 | 2 | 9/10 |
| Sunnu et al., 2022 [62] | 1 | 0 | 2 | 2 | 0 | 5/10 |
| Temiz et al., 2020 [63] | 2 | 0 | 2 | 2 | 1 | 7/10 |
| Vahidi et al., 2022 [64] | 0 | 0 | 2 | 2 | 0 | 4/10 |
| Wang et al., 2022 [65] | 0 | 0 | 2 | 2 | 1 | 5/10 |
| Xu et al., 2021 [66] | 0 | 1 | 2 | 2 | 2 | 7/10 |

However, when compare, we found no significant variability between sperm concentration before and after SARS-CoV-2 treatment (SMD -0.21 [95% CI: -0.53, 0.10] $P = 0.19$) and there was a significant inter-study diversity ($I^2 = 69\%$; $X^2$ $P = 0.001$), even after sensitivity analysis (SMD -0.18 [95% CI: -0.59, 0.23] $P = 0.39$), and there was no marked heterogeneity between studies ($I^2 = 67\%$; $X^2$ $P = 0.010$) (Fig 6B). Notwithstanding, SARS-CoV-2 significantly reduced

**Table 3.** Risk of bias assessment of the eligible studies.

| Study | Selection of exposed cohort | Selection of non-exposed cohort | Assessment of exposure | Demonstration of outcome | Comparability (basics) | Comparability (others) | Assessment outcome | Length of follow-up | Adequacy of follow-up | Total |
|---|---|---|---|---|---|---|---|---|---|---|
| Abbas et al., 2022 [27] | 1 | 1 | 0 | 1 | 1 | 0 | 1 | 0 | 0 | 5/9 |
| Aksak et al., 2022 [28] | 1 | 1 | 1 | 1 | 1 | 1 | 1 | 0 | 0 | 7/9 |
| Al-Alami et al. 2022 [29] | 1 | 1 | 1 | 1 | 0 | 0 | 1 | 0 | 0 | 5/9 |
| Al-Bashiti et al, 2022 [30] | 1 | 1 | 1 | 1 | 1 | 0 | 1 | 0 | 0 | 6/9 |
| Azzawi and Abdulrahman, 2022 [31] | 1 | 1 | 1 | 1 | 1 | 0 | 1 | 0 | 0 | 6/9 |
| Best et al, 2021 [32] | 1 | 1 | 1 | 1 | 1 | 1 | 1 | 1 | 1 | 9/9 |
| Camici et al., 2021 [33] | 1 | 1 | 1 | 1 | 1 | 1 | 1 | 0 | 0 | 6/9 |
| Cinislioglu et al., 2022 | 1 | 1 | 1 | 1 | 1 | 1 | 1 | 1 | 1 | 9/9 |
| Dipankar et al., 2022 [35] | 1 | 0 | 1 | 1 | 1 | 1 | 1 | 1 | 1 | 8/9 |
| Enikeev et al., 2022 [36] | 1 | 1 | 1 | 1 | 1 | 1 | 1 | 1 | 1 | 9/9 |
| Erbay et al, 2021 [37] | 1 | 0 | 1 | 1 | 1 | 1 | 1 | 0 | 0 | 6/9 |
| Falahieh et al., 2021 [38] | 1 | 0 | 1 | 1 | 1 | 1 | 1 | 1 | 1 | 8/9 |
| Gacci et al., 2021 [39] | 1 | 0 | 1 | 1 | 1 | 1 | 1 | 1 | 1 | 8/9 |
| Gul et al., 2021 [40] | 1 | 1 | 1 | 1 | 1 | 1 | 1 | 0 | 0 | 7/9 |
| Guo et al., 2021 [41] | 1 | 1 | 1 | 1 | 1 | 0 | 1 | 1 | 1 | 8/9 |
| Hadisi et al., 2022 [42] | 1 | 1 | 1 | 1 | 1 | 1 | 1 | 0 | 0 | 7/9 |
| Hamarat et al., 2022 [43] | 1 | 0 | 1 | 1 | 1 | 0 | 1 | 1 | 1 | 7/9 |
| Holtmann et al., 2020 [44] | 1 | 1 | 1 | 1 | 1 | 0 | 1 | 0 | 0 | 6/9 |
| Hu et al., 2022 [45] | 1 | 1 | 1 | 1 | 1 | 0 | 1 | 1 | 1 | 8/9 |
| Kadihasanoglu et al., 2021 [46] | 1 | 1 | 1 | 1 | 1 | 1 | 1 | - | - | 7/9 |
| Karkin and Gürlen, 2022 [47] | 1 | 0 | 1 | 1 | 1 | 1 | 1 | 1 | 1 | 8/9 |
| Koç and Keseroğlu, 2021 [48] | 1 | 0 | 1 | 1 | 0 | 0 | 1 | 0 | 0 | 4/9 |
| Kumar et al., 2023 [49] | 1 | 1 | 1 | 1 | 1 | 0 | 1 | 0 | 0 | 6/9 |
| Li et al., 2020 [50] | 1 | 1 | 1 | 1 | 1 | 1 | 1 | 0 | 0 | 7/9 |
| Livingstone et al., 2022 [51] | 1 | 1 | 1 | 1 | 1 | 0 | 1 | 0 | 0 | 6/9 |

*(Continued)*

**Table 3.** (Continued)

| Study | Selection of exposed cohort | Selection of non-exposed cohort | Assessment of exposure | Demonstration of outcome | Comparability (basics) | Comparability (others) | Assessment outcome | Length of follow-up | Adequacy of follow-up | Total |
|---|---|---|---|---|---|---|---|---|---|---|
| Ma et al., 2020 [52] | 1 | 1 | 1 | 1 | 1 | 0 | 1 | 1 | 1 | 8/9 |
| Maleki and Tartibian, 2021 [53] | 1 | 1 | 1 | 1 | 1 | 0 | 1 | 1 | - | 7/9 |
| Okçelik, 2020 [54] | 1 | 1 | 1 | 1 | 1 | 1 | 1 | 0 | 0 | 7/9 |
| Paoli et al., 2023 [55] | 1 | 1 | 1 | 1 | 1 | 1 | 1 | 1 | 1 | 9/9 |
| Pazir et al., 2021 [56] | 1 | 0 | 1 | 1 | 1 | 1 | 1 | 0 | 0 | 6/9 |
| Piroozmanesh et al., 2021 [57] | 1 | 1 | 1 | 1 | 1 | 1 | 1 | 0 | 0 | 7/9 |
| Rafiee and Tabei, 2021 [58] | 1 | 1 | 1 | 1 | 1 | 1 | 1 | 1 | 1 | 9/9 |
| Ruan et al., 2021 [59] | 1 | 1 | 1 | 1 | 1 | 1 | 1 | 0 | 0 | 8/9 |
| Salonia et al., 2021 [60] | 1 | 1 | 1 | 1 | 1 | 1 | 1 | 0 | 0 | 7/9 |
| Salonia et al., 2022 [61] | 1 | 0 | 1 | 1 | 1 | 0 | 1 | 1 | 1 | 7/9 |
| Salonia et al., 2021 [60] | 1 | 1 | 1 | 1 | 1 | 0 | 1 | 0 | 0 | 6/9 |
| Sunnu et al., 2022 [62] | 1 | 0 | 1 | 1 | 1 | 0 | 1 | 1 | 1 | 7/9 |
| Temiz et al., 2020 [63] | 1 | 1 | 1 | 1 | 1 | 1 | 1 | 1 | 1 | 9/9 |
| Vahidi et al., 2022 [64] | 1 | 0 | 1 | 1 | 1 | 1 | 1 | 0 | 0 | 6/9 |
| Wang et al., 2022 [65] | 1 | 1 | 1 | 1 | 1 | 0 | 1 | 0 | 0 | 6/9 |
| Xu et al., 2021 [66] | 1 | 1 | 1 | 1 | 1 | 0 | 1 | 0 | 0 | 6/9 |

sperm concentration of the patients when compared with the pre-COVID period (SMD -0.42 [95% CI: -0.70, -0.14] $P = 0.004$), we found no marked heterogeneity between studies ($I^2 = 69\%$; $X^2 P = 0.002$). After sensitivity analysis, it was still observed that SARS-CoV-2 significantly reduced sperm concentration when compared with the pre-COVID values of the patients (SMD -0.31 [95% CI: -0.50, -0.12] $P = 0.001$), and there existed no significant inter-study variability ($I^2 = 32\%$; $X^2 P = 0.21$) (Fig 6C). The publication bias as depicted by the funnels' plots are shown in Fig 7.

**Sperm viability.** SARS-CoV-2 significantly lowered sperm viability in comparison to SARS-CoV-2 uninfected individuals (SMD -1.08 [95% CI: -1.83, -0.33] $P = 0.005$). There was a notable inter-study diversity ($I^2 = 88\%$; $X^2 P < 0.00001$). Sensitivity analysis demonstrated that SARS-CoV-2 yet significantly reduced sperm viability when compared to the control (SMD -1.34 [95% CI: -1.95, -0.72] $P < 0.0001$), and there was a substantial inter-study diversity ($I^2 = 73\%$; $X^2 P = 0.01$) (Fig 8A). Moreover, sperm viability was significantly dropped in SARS-CoV-2 positive individuals before treatment in comparison to after treatment (SMD -0.84

**Table 4. Assessment of certainty of evidence of the eligible studies.**

| Study | Initial rating | Downgrading? | Upgrading? | Confidence in body of evidence |
|---|---|---|---|---|
| Abbas et al., 2022 [27] | High | Yes↓ | No | Moderate |
| Aksak et al., 2022 [28] | High | No | No | High |
| Al-Alami et al., 2022 [29] | Moderate | Yes, 1 | Yes, 1 | Low |
| Al-Bashiti et al., 2022 [30] | High | Yes, 1 | No | Moderate |
| Azzawi and Abdulrahman, 2022 [31] | High | Yes, 1 | No | Moderate |
| Best et al, 2021 [32] | High | No | No | High |
| Camici et al., 2021 [33] | High | No | No | High |
| Cinislioglu et al., 2022 [34] | High | No | No | High |
| Dipankar et al., 2022 [35] | Moderate | No | No | Moderate |
| Enikeev et al., 2022 [36] | High | No | No | High |
| Erbay et al., 2021 [37] | Moderate | Yes, 1 | Yes, 1 | Moderate |
| Falahieh et al., 2021 [38] | Moderate | No | No | Moderate |
| Gacci et al., 2021 [39] | High | yes, 1 | No | Moderate |
| Gul et al., 2021 [40] | High | Yes (2) | No | Moderate |
| Guo et al., 2021 [41] | High | Yes, 1 | No | Moderate |
| Hadisi et al., 2022 [42] | High | Yes, 1 | No | Moderate |
| Hamarat et al., 2022 [43] | High | No | No | High |
| Holtmann et al., 2020 [44] | High | Yes, 1 | No | Moderate |
| Hu et al., 2022 [45] | High | No | No | High |
| Kadihasanoglu et al., 2021 [46] | High | Yes, 1 | Yes | High |
| Karkin and Gürlen, 2022 [47] | Moderate | Yes, 1 | No | High |
| Koç & Keseroğlu, 2021 [48] | Moderate | Yes,1 | No | Low |
| Kumar et al., 2023 [49] | Moderate | No | Yes, 1 | High' |
| Li et al., 2020 [50] | High | No | No | High |
| Livingstone et al., 2022 [51] | High | Yes, 1 | No | Moderate |
| Ma et al., 2021 [52] | High | Yes, 1 | Yes, 1 | High |
| Maleki and Tartibian, 2021 [53] | High | No | No | High |
| Okçelik, 2020 [54] | High | Yes, 1 | No | Moderate |
| Paoli et al., 2023 [55] | Moderate | No | Yes, 1 | High |
| Pazir et al., 2021 [56] | Moderate | Yes, 1 | No | Low |
| Piroozmanesh et al., 2021 [57] | High | No | No | High |
| Rafiee and Tabei, 2021 [58] | High | No | No | High |
| Ruan et al., 2021 [59] | High | No | No | High |
| Salonia et al., 2021 [60] | High | Yes, 1 | Yes, 1 | High |
| Salonia et al., 2022 [61] | High | Yes, 1 | Yes, 1 | High |
| Sunnu et al., 2022 [62] | Moderate | No | No | Moderate |
| Temiz et al., 2020 [63] | High | No | No | High |
| Vahidi et al 2022 [64] | High | Yes, 1 | No | Moderate |
| Wang et al 2022 [65] | High | Yes, 1 | No | Moderate |
| Xu et al., 2021 [66] | High | Yes (1) | N0 | Moderate |

[95% CI: -1.37, -0.31] $P = 0.002$), and there was a significant inter-study diversity ($I^2 = 75\%$; $X^2$ $P = 0.003$). After sensitivity analysis, there was a significant drop in sperm viability among patients infected with SARS-CoV-2 positive when juxtaposed with the control (SMD -0.53 [95% CI: -0.86, -0.20] $P = 0.002$), but there existed no significant inter-study variability ($I^2 = 0\%$; $X^2$ $P = 0.53$) (Fig 8B). In addition, when colligated with their premorbid state, sperm viability was significantly reduced in SARS-CoV-2 positive patients (SMD -0.85 [95% CI: -1.43,

**A.**

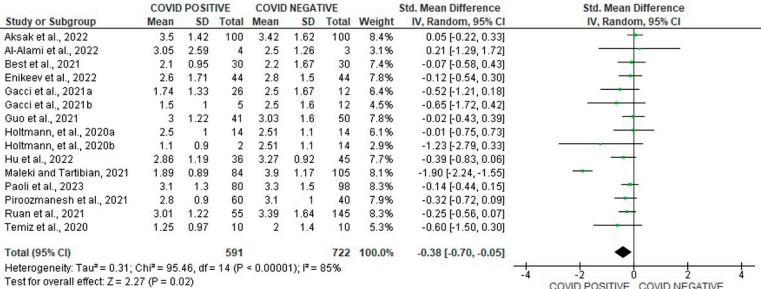

**Sensitivity analysis**

**B.**

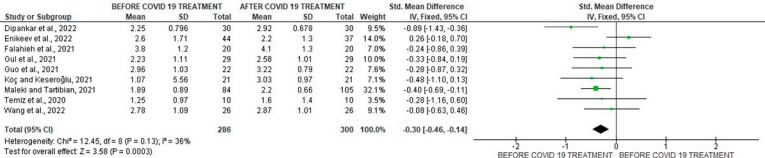

**Sensitivity analysis**

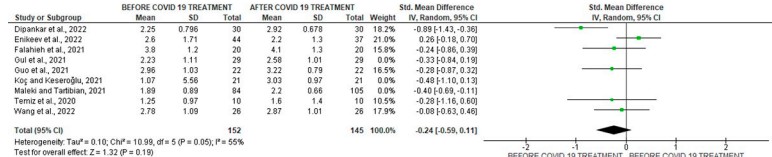

**C.**

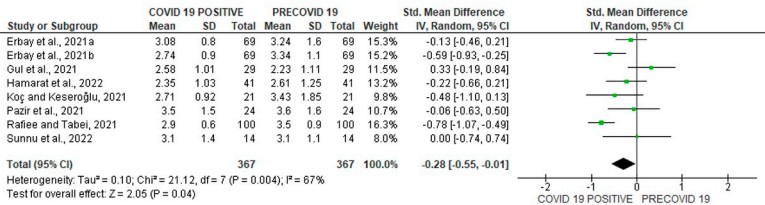

**Sensitivity analysis**

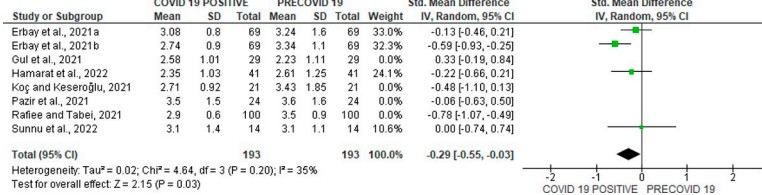

**Fig 2.** Forest plot of ejaculate volume comparing between COVID-19 positive and COVID-19 negative patients (A), before COVID-19 treatment and after COVID-19 treatment (B), and COVID-19 positive and preCOVID-19 period (C).

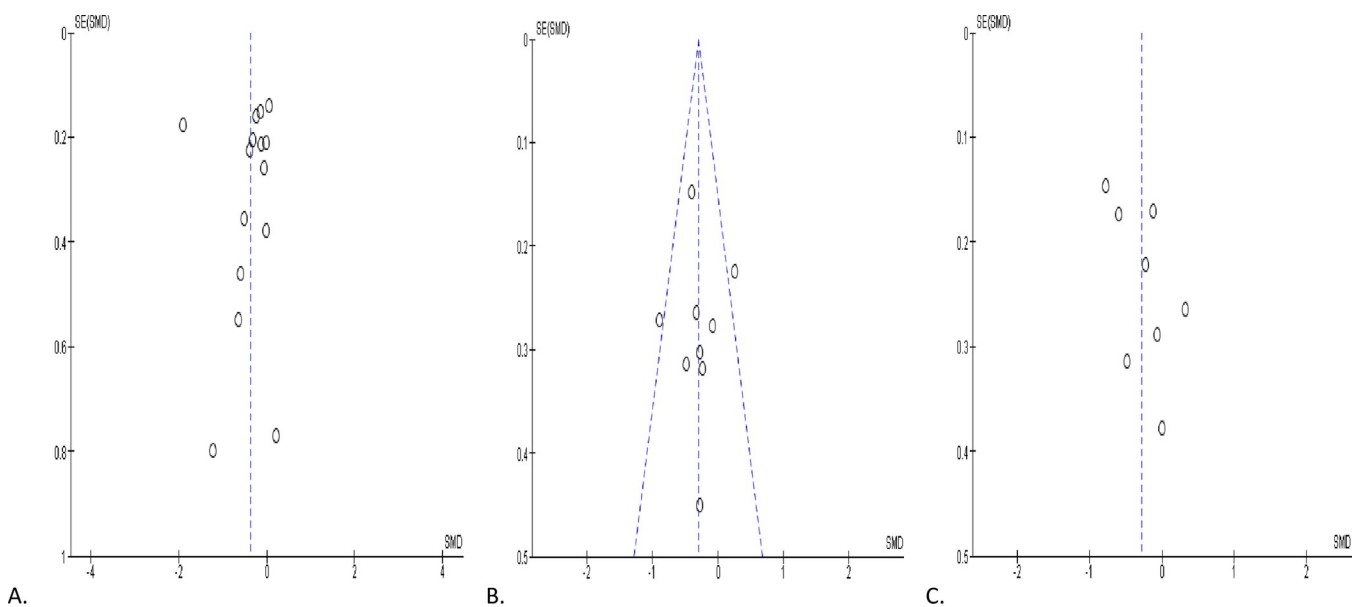

**Fig 3.** Funnel plot of ejaculate volume comparing between COVID-19 positive and COVID-19 negative patients (A), before COVID-19 treatment and after COVID-19 treatment (B), and COVID-19 positive and preCOVID-19 period (C).

-0.26] $P$ = 0.005). There was a substantial heterogeneity between studies ($I^2$ = 82%; $X^2$ $P$ = 0.02) (Fig 8C). Fig 9 shows the funnels' plots demonstrating the publication bias.

**Total and progressive sperm motility.** The total sperm motility was only marginally diminished in SARS-CoV-2 positive patients when compared with the control (SMD -0.30 [95% CI: -0.61, 0.00] $P$ = 0.05), and there was a marked heterogeneity between studies ($I^2$ = 63%; $X^2$ $P$ = 0.008). After sensitivity analysis, the difference in the total sperm motility remained insignificant (SMD -0.34 [95% CI: -0.86, 0.18] $P$ = 0.20), and there was a marked heterogeneity between studies ($I^2$ = 82%; $X^2$ $P$ < 0.0001) (Fig 10A). Also, there was a marginal decline in total sperm motility in SARS-CoV-2 positive patients before, juxtaposed with after treatment (SMD -0.34 [95% CI: -0.86, 0.18] $P$ = 0.20), and there was a marked heterogeneity between studies ($I^2$ = 82%; $X^2$ $P$ < 0.0001), even after sensitivity analysis (SMD -0.54 [95% CI: -1.36, 0.28] $P$ = 0.20), and there was a marked heterogeneity between studies ($I^2$ = 84%; $X^2$ $P$ = 0.0002) (Fig 10B). However, SARS-CoV-2 led to a marked decline in total sperm motility in infected patients when compared with their premorbid values (SMD -0.68 [95% CI: -1.12, -0.24] $P$ = 0.002), and there was a marked heterogeneity between studies ($I^2$ = 87%; $X^2$ $P$ < 0.00001). After sensitivity analysis, the significant difference in total sperm motility persisted in SARS-CoV-2 positive patients between the infected state and premorbid state (SMD -0.73 [95% CI: -1.42, -0.04] $P$ = 0.04), and there was a significant inter-study diversity ($I^2$ = 90%; $X^2$ $P$ < 0.00001) (Fig 10C). The funnels' plots showing the publication bias are presented in Fig 11.

When colligated with the controls, progressive sperm motility substantially diminished in SARS-CoV-2 positive patients (SMD -0.48 [95% CI: -0.94, -0.02] $P$ = 0.04), and there was a marked heterogeneity between studies ($I^2$ = 86%; $X^2$ $P$ < 0.00001); although after sensitivity analysis, SARS-CoV-2 only caused a marginal decline in progressive sperm motility when compared with the control (SMD -0.51 [95% CI: -1.09, 0.07] $P$ = 0.08), and there was a marked heterogeneity between studies ($I^2$ = 89%; $X^2$ $P$ < 0.00001) (Fig 12A). In addition, COVID-19 significantly reduced progressive sperm motility in infected patients before treatment when compared with after treatment (SMD -0.41 [95% CI: -0.77, -0.05] $P$ = 0.02), and there was a

**A.**

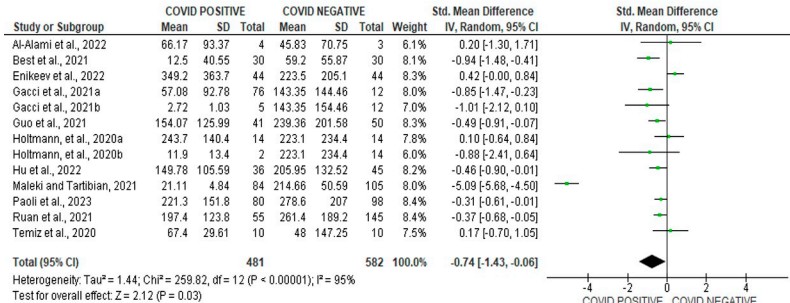

**Sensitivity analysis**

**B.**

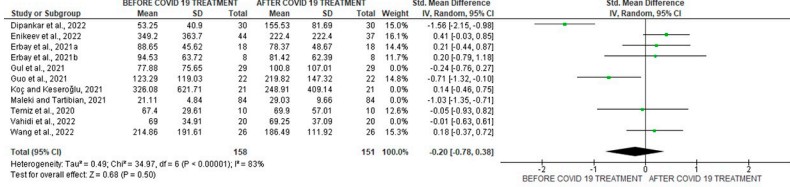

**Sensitivity analysis**

**C.**

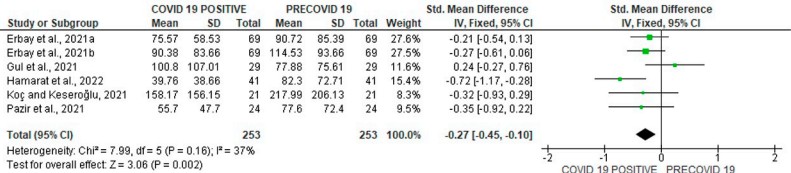

**Fig 4.** Forest plot of sperm count comparing between COVID-19 positive and COVID-19 negative patients (A), before COVID-19 treatment and after COVID-19 treatment (B), and COVID-19 positive and preCOVID-19 period (C).

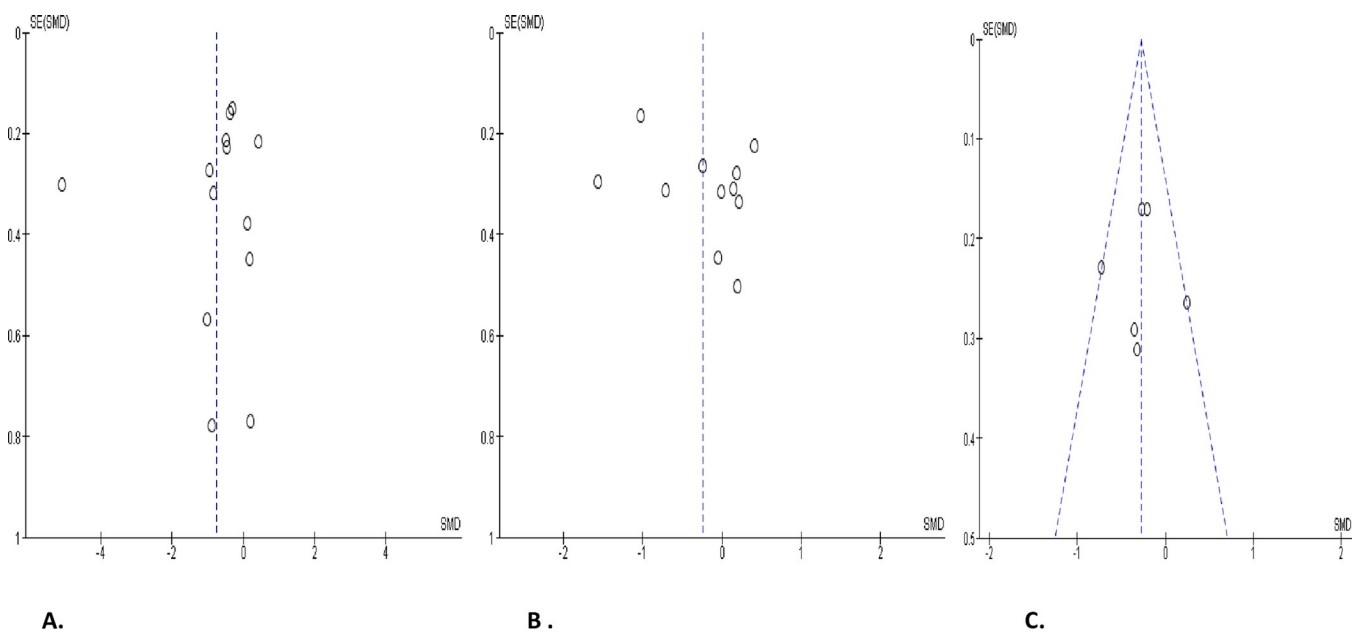

**Fig 5.** Funnel plot of sperm count comparing between COVID-19 positive and COVID-19 negative patients (A), before COVID-19 treatment and after COVID-19 treatment (B), and COVID-19 positive and preCOVID-19 period (C).

significant inter-study diversity ($I^2$ = 77%; $X^2$ $P$ < 0.0001). Following sensitivity analysis, it was revealed that SARS-CoV-2 significantly reduced progressive sperm motility in infected patients before treatment when compared with after treatment (SMD -0.53 [95% CI: -1.02, -0.05] $P$ = 0.03), and there was a marked heterogeneity between studies ($I^2$ = 74%; $X^2$ $P$ = 0.002) (Fig 12B). Furthermore, SARS-CoV-2 caused a significant decline in progressive sperm motility in infected cohorts when compared with their premorbid state (SMD -0.49 [95% CI: -0.80, -0.19] $P$ = 0.002), and there was a significant inter-study variation ($I^2$ = 65%; $X^2$ $P$ = 0.009); however, this was observed to be marginal after sensitivity analysis (SMD -0.18 [95% CI: -0.56, 0.19] $P$ = 0.34), and there was no significant inter-study diversity ($I^2$ = 0%; $X^2$ $P$ = 0.81) (Fig 12C). The funnels' plots showing publication bias are presented in Fig 13.

**Sperm morphology.** SARS-CoV-2 infection did not significantly alter normal sperm morphology when compared with the COVID-19-negative controls (SMD -0.49 [95% CI: -1.33, 0.34] $P$ = 0.25), and there was a marked heterogeneity between studies ($I^2$ = 95%; $X^2$ $P$ < 0.00001), even after sensitivity analysis (SMD -0.70 [95% CI: -1.83, 0.43] $P$ = 0.23), and there was a significant inter-study variation ($I^2$ = 96%; $X^2$ $P$ < 0.00001) (Fig 14A). Similarly, SARS-CoV-2 did not considerably affect sperm morphology in infected patients before treatment in comparison with after treatment (SMD -0.19 [95% CI: -0.58, 0.21] $P$ = 0.36), and there was a marked heterogeneity between studies ($I^2$ = 84%; $X^2$ $P$ < 0.00001), despite sensitivity analysis (SMD -0.25 [95% CI: -0.81, 0.31] $P$ = 0.38), and there was a marked heterogeneity between studies ($I^2$ = 85%; $X^2$ $P$ < 0.00001) (Fig 14B). More so, SARS-CoV-2 caused a decline in normal sperm morphology in infected cohorts when colligated with their pre-morbid states (SMD -0.83 [95% CI: -1.69, 0.03] $P$ = 0.06), and there was a marked heterogeneity between studies ($I^2$ = 92%; $X^2$ $P$ < 0.00001). Nevertheless, there was a substantial reduction in the proportion of sperm with normal morphology after sensitivity analysis in SARS-CoV-2 positive patients when juxtaposed with their pre-COVID states (SMD -0.65 [95% CI: -1.03, -0.26] $P$ = 0.0010), and there was no marked heterogeneity between studies ($I^2$ = 0%; $X^2$ $P$ = 0.50) (Fig 14C). The publication bias as depicted by funnels' plots are presented in Fig 15.

**A.**

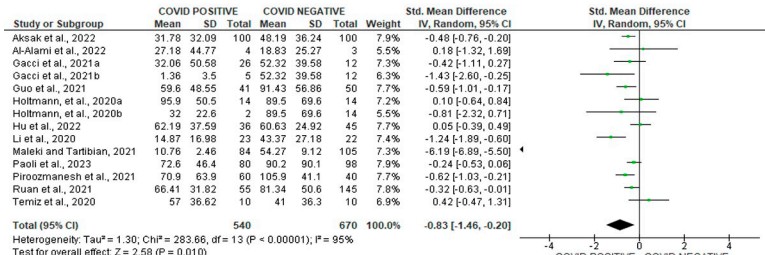

**Sensitivity analysis**

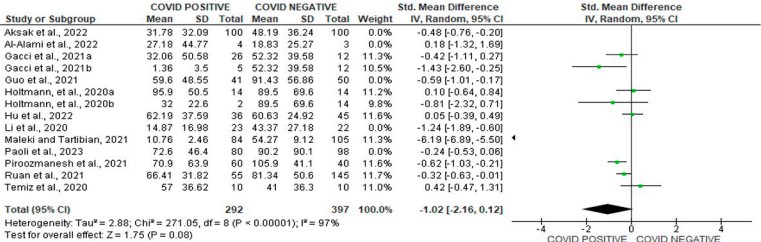

**B.**

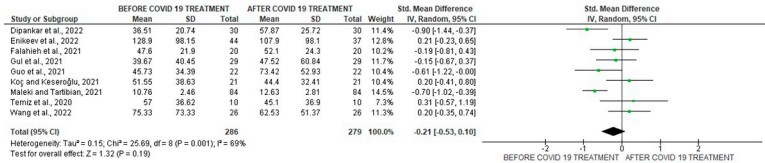

**Sensitivity analysis**

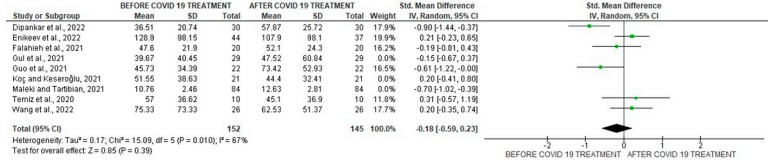

**C.**

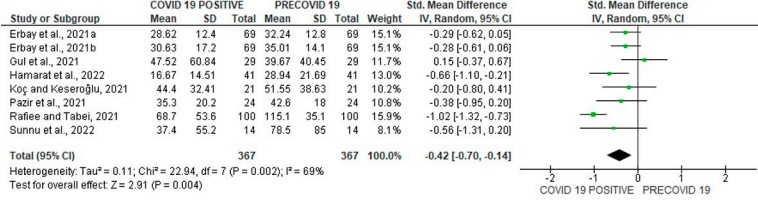

**Sensitivity analysis**

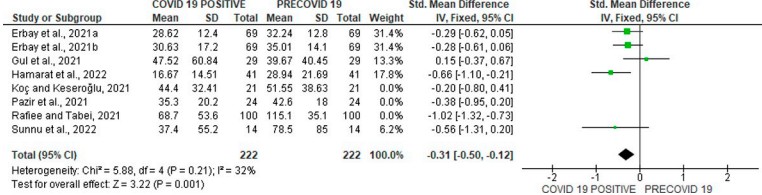

**Fig 6.** Forest plot of sperm concentration comparing between COVID-19 positive and COVID-19 negative patients (A), before COVID-19 treatment and after COVID-19 treatment (B), and COVID-19 positive and preCOVID-19 period (C).

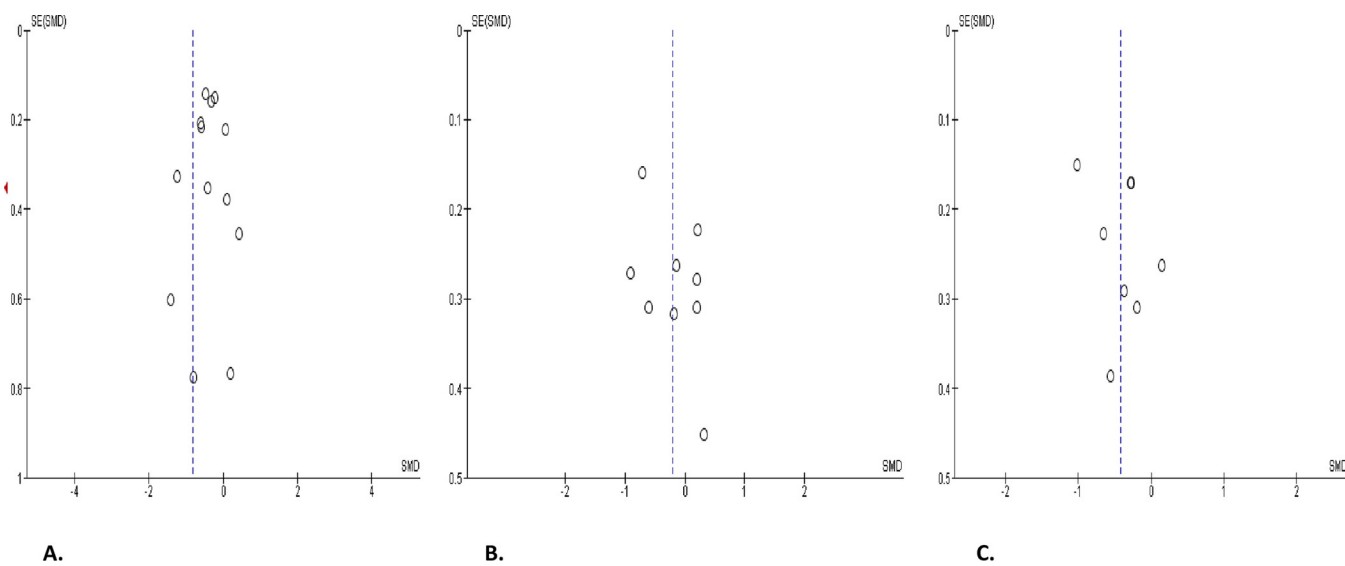

**Fig 7.** Funnel plot of sperm concentration comparing between COVID-19 positive and COVID-19 negative patients (A), before COVID-19 treatment and after COVID-19 treatment (B), and COVID-19 positive and preCOVID-19 period (C).

**Seminal leukocyte count.** Only two studies reported data on seminal fluid leukocyte, comparing COVID-positive and COVID-negative patients, while three studies reported these parameters comparing COVID-pre- and post- treatment status of the infected patients. Unexpectedly, SARS-CoV-2 infection did not alter seminal leukocyte levels when compared with controls (SMD -0.01 [95% CI: -0.46, 0.43] $P$ = 0.95), and there was no marked heterogeneity between studies ($I^2$ = 29%; $X^2$ $P$ = 0.24). In addition, when seminal leukocytes in SARS-CoV-2 positive patients were colligated before and after treatment, there was no marked heterogeneity (SMD 0.34 [95% CI: -0.33, 1.00] $P$ = 0.32), and there was a marked heterogeneity between studies ($I^2$ = 80%; $X^2$ $P$ = 0.007) (Fig 16). The funnels' plots showing the publication bias are shown in Fig 17.

**Circulating testosterone, oestrogen, and prolactin levels.** SARS-CoV-2 infection engendered a substantial diminution in serum testosterone level when collocated with covid-19-negative controls (SMD -1.00 [95% CI: -1.49, -0.51] $P$< 0.0001), and there was a marked heterogeneity between studies ($I^2$ = 96%; $X^2$ $P$ < 0.00001) (Fig 18A). However, SARS-CoV-2 infection did not significantly cause a wane in serum testosterone level in infected patients in comparison before and after treatment (SMD -0.87 [95% CI: -1.90, 0.16] $P$ = 0.10), and there was a significant inter-study diversity ($I^2$ = 95%; $X^2$ $P$ < 0.00001). After sensitivity analysis, serum testosterone level did not also show notable distinction between SARS-CoV-2 positive patients before and after treatment (SMD -1.30 [95% CI: -3.27, 0.67] $P$ = 0.20), and there was a significant inter-study diversity ($I^2$ = 98%; $X^2$ $P$ < 0.00001) (Fig 18B). More so, circulating testosterone level was not significantly altered in SARS-CoV-2 positive patients in colligation with their premorbid states (SMD -0.51 [95% CI: -1.22, 0.19] $P$ = 0.15), and there was a marked heterogeneity between studies ($I^2$ = 88%; $X^2$ $P$ = 0.0003) (Fig 18C). The publication bias using funnels' plots are shown in Fig 19.

In addition, serum concentration of oestrogen was marginally higher in SARS-CoV-2 patients in comparison with uninfected controls (SMD 0.62 [95% CI: 0.18, 1.07] $P$ = 0.006). There was a marked heterogeneity between studies ($I^2$ = 70%; $X^2$ $P$ = 0.04) (Fig 20A). The funnel's plot showing the publication bias is shown in Fig 20B.

**A.**

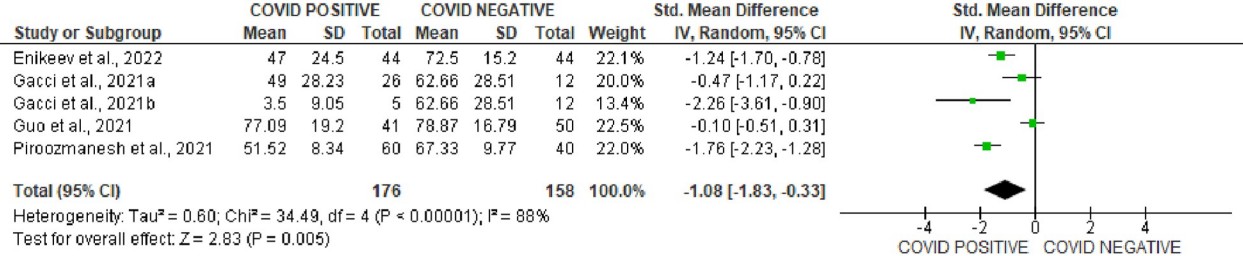

**Sensitivity analysis**

**B.**

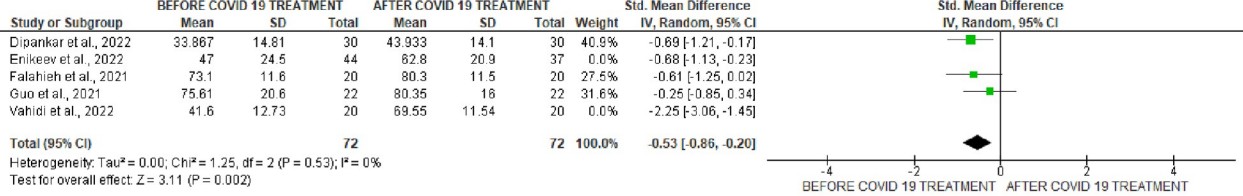

**Sensitivity analysis**

**C.**

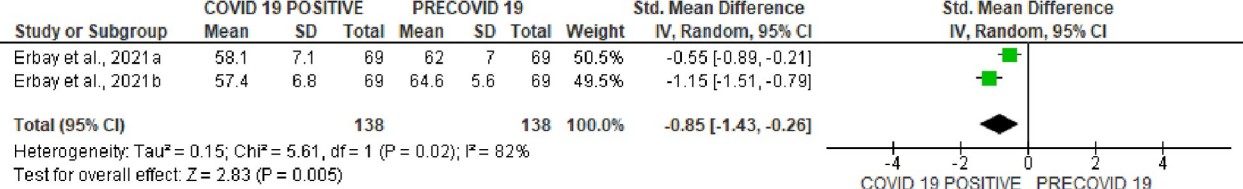

**Fig 8.** Forest plot of sperm viability comparing between COVID-19 positive and COVID-19 negative patients (A), before COVID-19 treatment and after COVID-19 treatment (B), and COVID-19 positive and preCOVID-19 period (C).

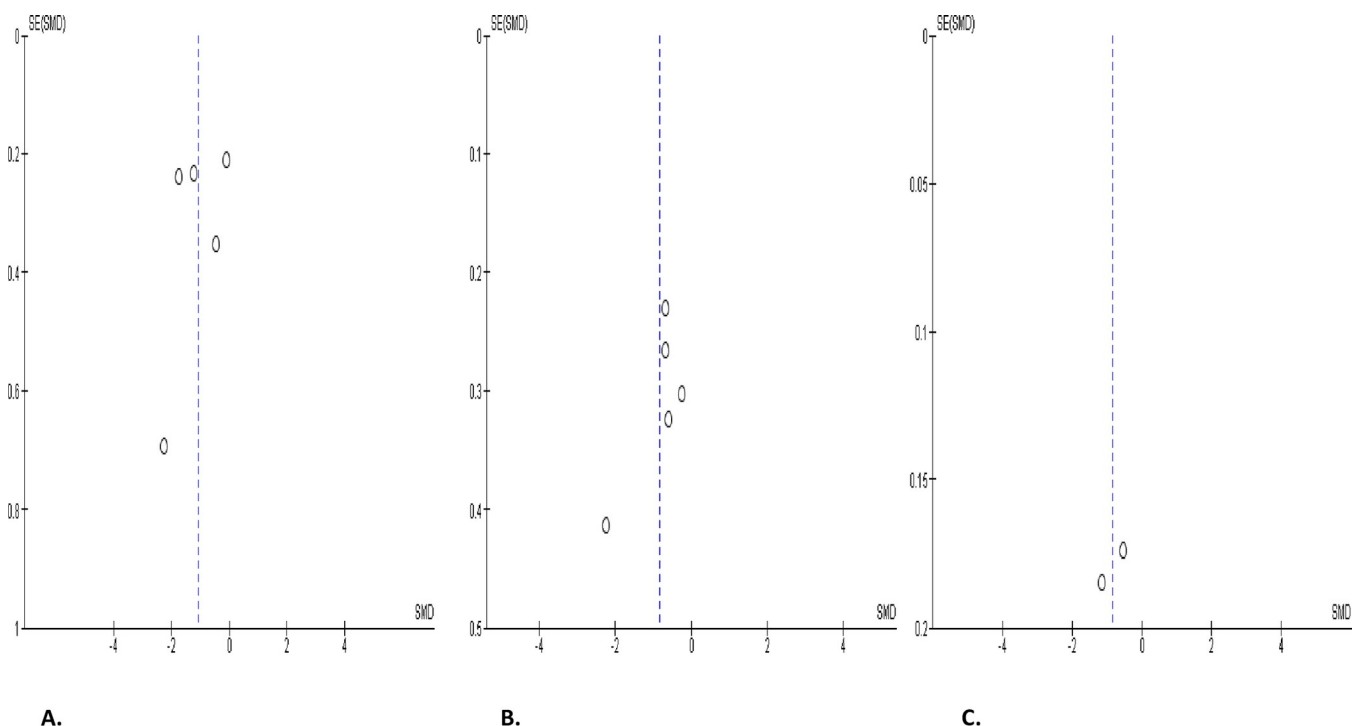

**Fig 9.** Funnel plot of sperm viability comparing between COVID-19 positive and COVID-19 negative patients (A), before COVID-19 treatment and after COVID-19 treatment (B), and COVID-19 positive and preCOVID-19 period (C).

However, SARS-CoV-2 infection significantly increased serum prolactin concentration when compared with uninfected control (SMD 0.53 [95% CI: 0.11, 0.95] $P$ = 0.01), and there was a notable heterogeneity between studies ($I^2$ = 86%; $X^2$ $P$ < 0.00001) (Fig 21A). In comparison with SARS-CoV-2 positive patients after treatment, SARS-CoV-2 infection did not significantly alter serum prolactin level (SMD 0.39 [95% CI: -0.85, 1.64] $P$ = 0.54), and there was a substantial inter-study variation ($I^2$ = 91%; $X^2$ $P$ < 0.0001) (Fig 21B). The funnels' plots showing the publication bias are shown in Fig 22.

**Serum levels of gonadotropins.** Serum level of LH was significantly elevated in SARS-CoV-2 positive when juxtaposed with the uninfected control (SMD 0.75 [95% CI: 0.19, 1.31] $P$ = 0.009), and there was a marked heterogeneity between studies ($I^2$ = 96%; $X^2$ $P$ < 0.0001). After sensitivity analysis, serum LH level remained higher in SARS-CoV-2 positive cohorts in colligation with the negative cohorts (SMD 1.09 [95% CI: 0.10, 2.07] $P$ = 0.03), and there was a substantial heterogeneity between studies ($I^2$ = 97%; $X^2$ $P$ < 0.0001) (Fig 23A). However, serum LH level was not significantly different in SARS-CoV-2 positive before and after treatment (SMD 0.05 [95% CI: -0.28, 0.37] $P$ = 0.78), and there was no significant inter-study diversity ($I^2$ = 0%; $X^2$ $P$ = 0.76) (Fig 23B). In addition, there was no notable variance in serum LH levels in SARS-CoV-2 positive patients when compared with their pre-COVID state (SMD 0.54 [95% CI: -0.47, 1.56] $P$ = 0.29), and there was a substantial heterogeneity between studies ($I^2$ = 94%; $X^2$ $P$ < 0.00001) (Fig 23C). The publication bias, using funnels' plots, are shown in Fig 24.

Serum FSH was marginally increased in SARS-CoV-2 positive patients when compared with the control (SMD 0.13 [95% CI: -0.16, 0.43] $P$ = 0.37), and there was a noteworthy heterogeneity between studies ($I^2$ = 90%; $X^2$ $P$ < 0.00001), which persisted even after sensitivity analysis (SMD 0.13 [95% CI: -0.25, 0.51] $P$ = 0.50), and there was a marked heterogeneity between

**A.**

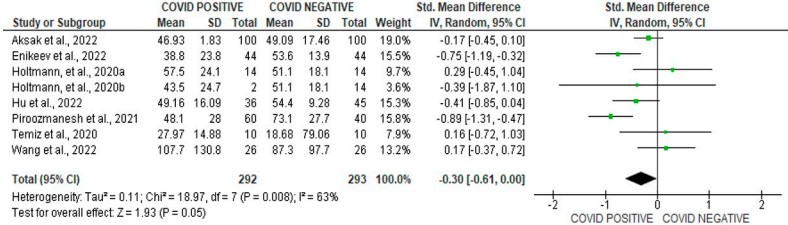

**B.**

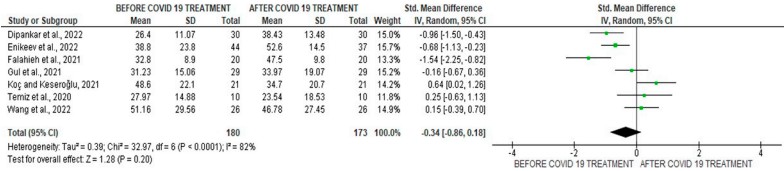

**C.**

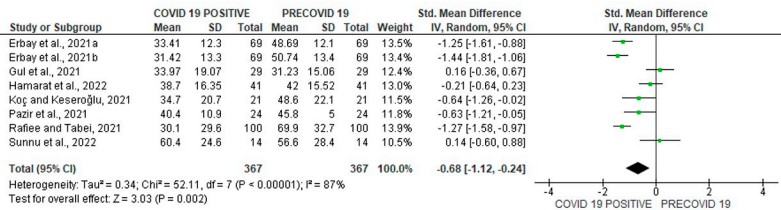

**Fig 10.** Forest plot of total sperm motility comparing between COVID-19 positive and COVID-19 negative patients (A), before COVID-19 treatment and after COVID-19 treatment (B), and COVID-19 positive and preCOVID-19 period (C).

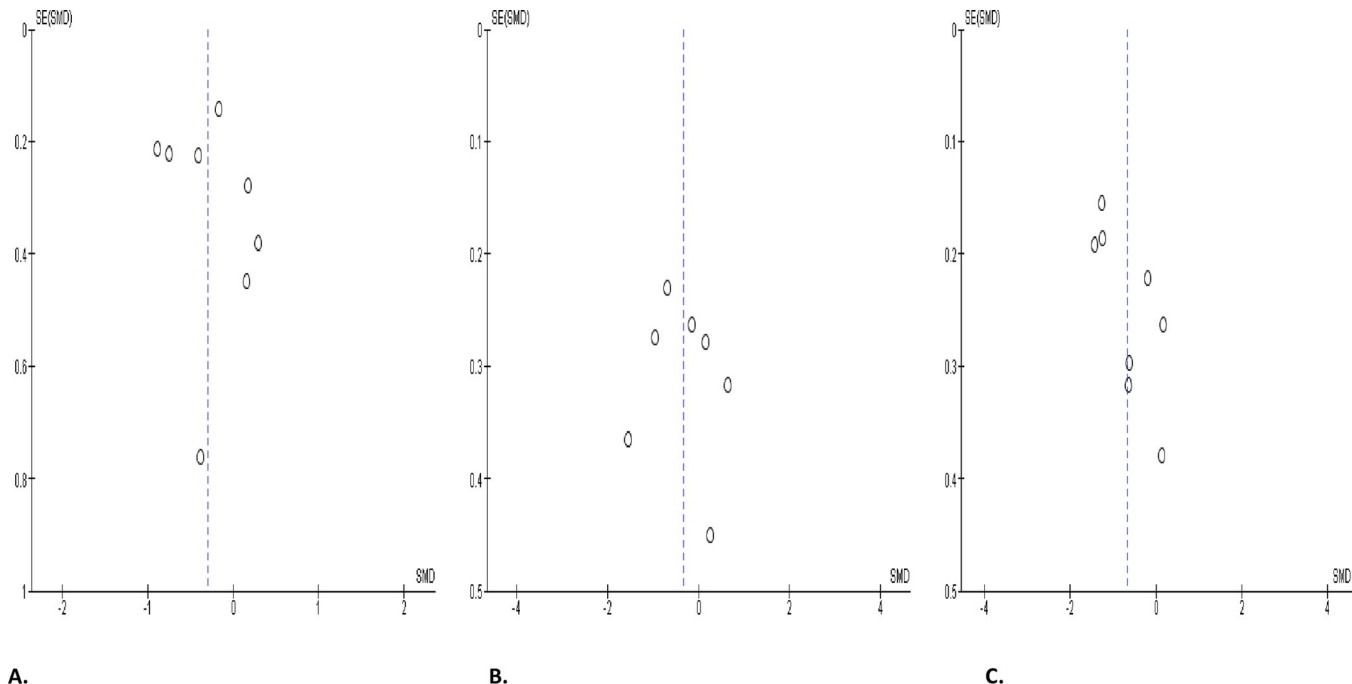

**A.**  **B.**  **C.**

**Fig 11.** Funnel plot of total sperm motility comparing between COVID-19 positive and COVID-19 negative patients (A), before COVID-19 treatment and after COVID-19 treatment (B), and COVID-19 positive and preCOVID-19 period (C).

studies ($I^2$ = 91%; $X^2$ $P$ < 0.00001) (Fig 25A). In comparison with infected patients after treatment, FSH level in infected patients was not significantly different (SMD -0.36 [95% CI: -1.07, 0.35] $P$ = 0.32), and there was a marked heterogeneity between studies ($I^2$ = 89%; $X^2$ $P$ < 0.0001) (Fig 25C). Also, FSH level did not show any significant difference in SARS-CoV-2 positive when compared with the preCOVID state (SMD 0.11 [95% CI: -0.03, 0.25] $P$ = 0.12), and there was no significant inter-study diversity ($I^2$ = 0%; $X^2$ $P$ = 0.98) (Fig 25C). The funnels' plot showing the publication bias are presented in Fig 26.

**Reproductive hormone indices.** Serum testosterone/LH and FSH/LH were compared in SARS-CoV-2 positive patients and the uninfected controls. It was observed that SARS-CoV-2 engendered a significant decline in testosterone/LH level when compared with the control (SMD -2.44 [95% CI: -3.69, -1.19] $P$ = 0.0001), and there existed a notable inter-study variation ($I^2$ = 99%; $X^2$ $P$ < 0.00001) (Fig 27A). The publication bias is shown in Fig 27B.

Furthermore, SARS-CoV-2 infection resulted in a marginal reduction in FSH/LH level when juxtaposed with the control (SMD -2.06 [95% CI: -4.36, 0.25] $P$ = 0.08), and there was a significant inter-study diversity ($I^2$ = 98%; $X^2$ $P$ < 0.00001) (Fig 28A). The publication bias is shown in Fig 28B.

## Discussion

Although the achievement of clinical pregnancy and live birth is the true test of infertility, conventional semen analysis remains the cornerstone of the diagnosis and management of male infertility [67]. Evaluation of male sex hormones is also a useful tool in the management of male infertility. Our present data revealed that SARS-CoV-2 caused reductions in ejaculate volume, sperm count, concentration, viability, normal morphology, and total and progressive motility. These findings were associated with SARS-CoV-2-induced decline in serum testosterone level, and increase in oestrogen, prolactin, LH, and testosterone/LH levels. These data

**A.**

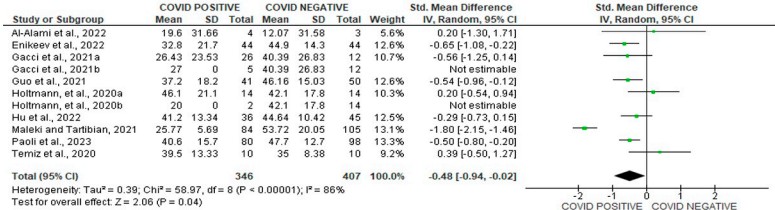

**Sensitivity analysis**

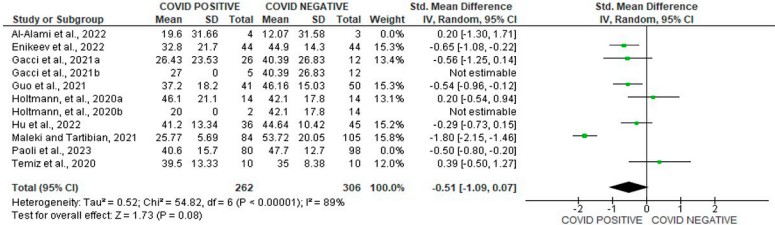

**B.**

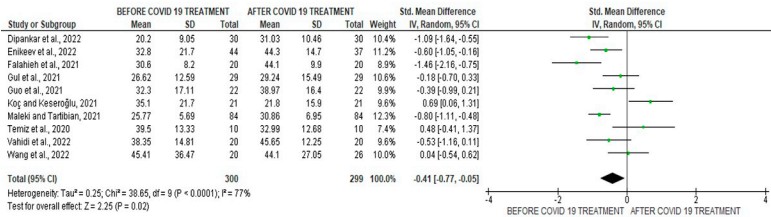

**Sensitivity analysis**

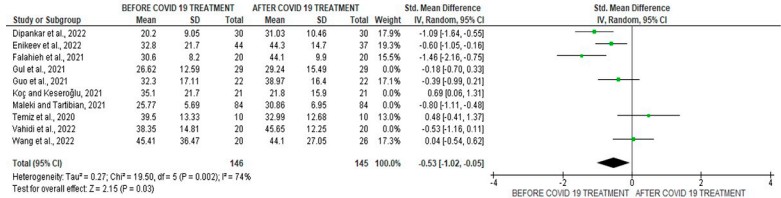

**C.**

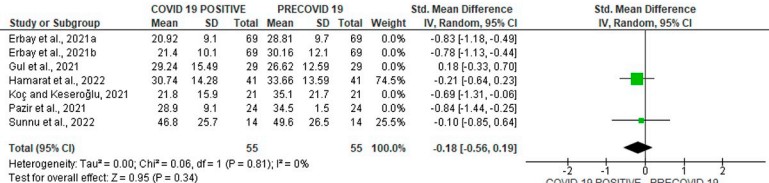

**Sensitivity analysis**

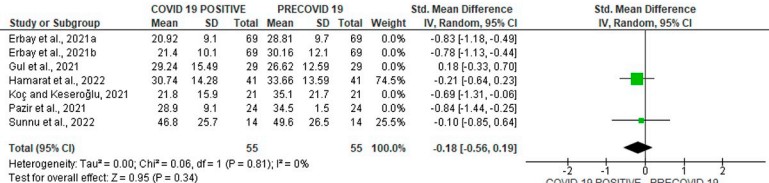

**Fig 12.** Forest plot of progressive sperm motility comparing between COVID-19 positive and COVID-19 negative patients (A), before COVID-19 treatment and after COVID-19 treatment (B), and COVID-19 positive and preCOVID-19 period (C).

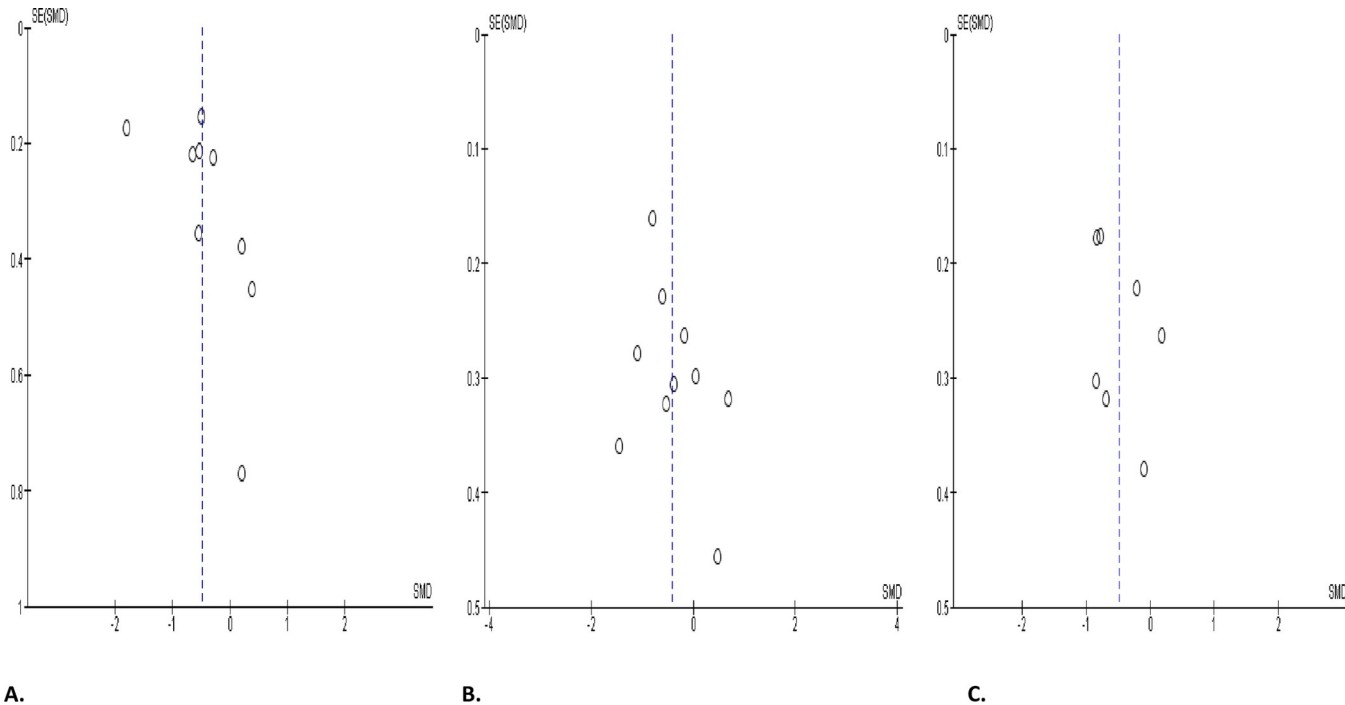

**A.**  **B.**  **C.**

**Fig 13.** Funnel plot of progressive sperm motility comparing between COVID-19 positive and COVID-19 negative patients (A), before COVID-19 treatment and after COVID-19 treatment (B), and COVID-19 positive and preCOVID-19 period (C).

convincingly demonstrate that SARS-CoV-2 may impede fertility in males by engendering a nadir of semen quality and distorting male reproductive hormone milieu.

The present findings corroborate and form an extension of the previous findings of the meta-analysis of Corona et al. [21], Tiwari et al. [22], and Xie et al [68]. Our present findings provide an update and robust data demonstrating the detrimental sequelae of SARS-CoV-2 on semen quality and male sex hormones. These data also augment the evidence available in the scientific literature that support the grievous consequence which SARS-CoV-2 impacts on male reproductive function.

It is plausible to infer that SARAS-CoV-2 may impair male fertility through multiple pathways. The expression of SARS-CoV-2 virus in the semen of infected patients [69–71] suggests that the virus may exert a local effect on the sperm cells. SARS-CoV-2 virus promotes oxidative stress evinced by heightened reactive oxygen species (ROS) generation, malondialdehyde (MDA) level and decline in total antioxidant capacity (TAC) in the semen fluid of infected patients [38]. Since the sperm cells are rich in polyunsaturated fatty acids that make them highly susceptible to ROS attack, SARS-CoV-2-induced ROS generation in the spermatozoa may cause oxidative sperm damage, leading to reduced sperm count, viability, motility, concentration, and normal morphology.

In addition, studies have shown that SARS-CoV-2 positively modulates cytokines[30] through extracellular-regulated protein kinase (ERK) and p38 mitogen-activated protein kinases (MAPK) activation [3,4,72], thus activating a cascade of immune responses, which lead to a hyper-inflammatory state that compromise the blood-testis-barrier [3,73,74] and increase the susceptibility of the testis and germ cells to SARS-CoV-2-driven ROS attack. This may explain the reduced semen quality and testosterone levels observed in SARS-CoV-2 positive patient. Since LH and FSH levels were not reduced in association with reduced testosterone, it is

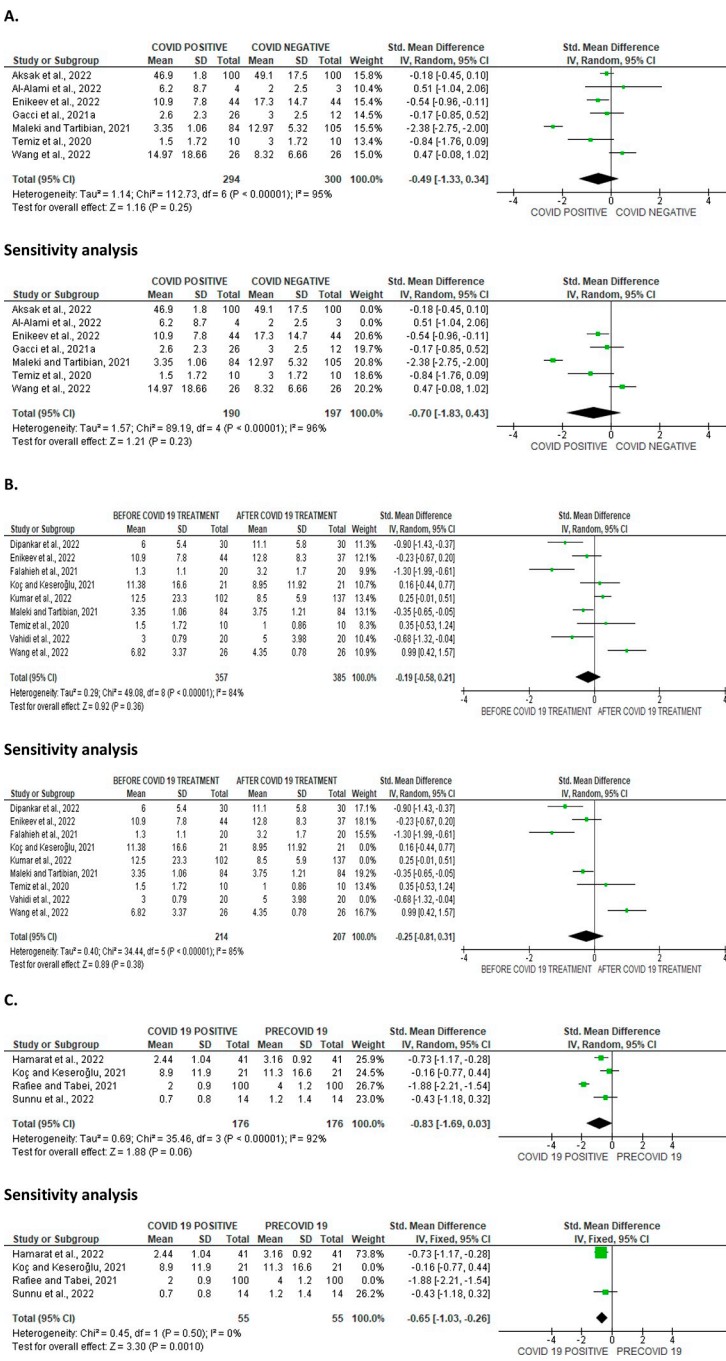

**Fig 14.** Forest plot of normal sperm morphology comparing between COVID-19 positive and COVID-19 negative patients (A), before COVID-19 treatment and after COVID-19 treatment (B), and COVID-19 positive and preCOVID-19 period (C).

credible to infer that SARS-CoV-2-induced testosterone decline is a local effect and not due to the suppression of the hypothalamic-pituitary-testicular axis. The observed rise in circulating oestrogen and prolactin concentrations in SARS-CoV-2 positive patients may also suggest the endocrine-disrupting activity of the viral infection as a pathway of impairing male fertility.

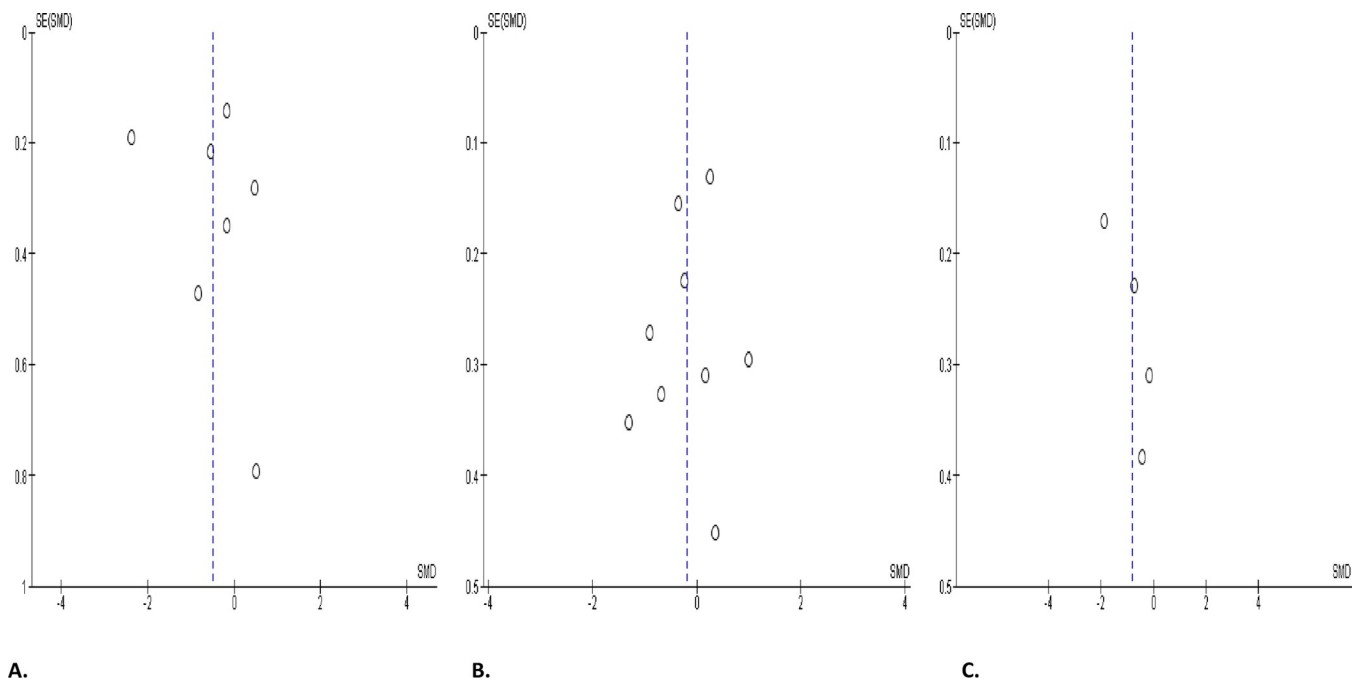

**Fig 15.** Funnel plot of normal sperm morphology comparing between COVID-19 positive and COVID-19 negative patients (A), before COVID-19 treatment and after COVID-19 treatment (B), and COVID-19 positive and preCOVID-19 period (C).

Beyond semen quality, SARS-CoV-2 infection may also impact on the success of testicular sperm extraction, hence on the outcome of assisted reproductive techniques (ART). Testosterone/LH is a known predictor of sperm concentration and successful sperm retrieval [75,76]; therefore, the reduced testosterone/LH level in SARS-CoV-infected patients explains the reduced sperm concentration found in the patients and also reveals a likelihood of reduced

**A.**

| Study or Subgroup | COVID POSITIVE Mean | SD | Total | COVID NEGATIVE Mean | SD | Total | Weight | Std. Mean Difference IV, Random, 95% CI |
|---|---|---|---|---|---|---|---|---|
| Enikeev et al., 2022 | 0.3 | 0.2 | 44 | 0.3 | 0.4 | 44 | 55.0% | 0.00 [-0.42, 0.42] |
| Gacci et al., 2021a | 0.03 | 0.07 | 26 | 0.08 | 0.2 | 12 | 30.1% | -0.39 [-1.08, 0.30] |
| Gacci et al., 2021b | 0.7 | 1.6 | 5 | 0.08 | 0.2 | 12 | 14.9% | 0.70 [-0.38, 1.77] |
| **Total (95% CI)** | | | 75 | | | 68 | 100.0% | -0.01 [-0.46, 0.43] |

Heterogeneity: Tau² = 0.05; Chi² = 2.83, df = 2 (P = 0.24); I² = 29%
Test for overall effect: Z = 0.06 (P = 0.95)

**B.**

| Study or Subgroup | BEFORE COVID 19 TREATMENT Mean | SD | Total | AFTER COVID 19 TREATMENT Mean | SD | Total | Weight | Std. Mean Difference IV, Random, 95% CI |
|---|---|---|---|---|---|---|---|---|
| Dipankar et al., 2022 | 9.5 | 5.8 | 30 | 6.7 | 3.3 | 30 | 33.6% | 0.59 [0.07, 1.10] |
| Enikeev et al., 2022 | 0.3 | 0.2 | 44 | 0.4 | 0.5 | 44 | 36.1% | -0.26 [-0.68, 0.16] |
| Falahieh et al., 2021 | 1.5 | 1.1 | 20 | 0.8 | 0.6 | 20 | 30.2% | 0.77 [0.13, 1.42] |
| **Total (95% CI)** | | | 94 | | | 94 | 100.0% | 0.34 [-0.33, 1.00] |

Heterogeneity: Tau² = 0.27; Chi² = 9.79, df = 2 (P = 0.007); I² = 80%
Test for overall effect: Z = 0.99 (P = 0.32)

**Fig 16.** Forest plot of seminal leukocyte count comparing between COVID-19 positive and COVID-19 negative patients (A) and before COVID-19 treatment and after COVID-19 treatment (B).

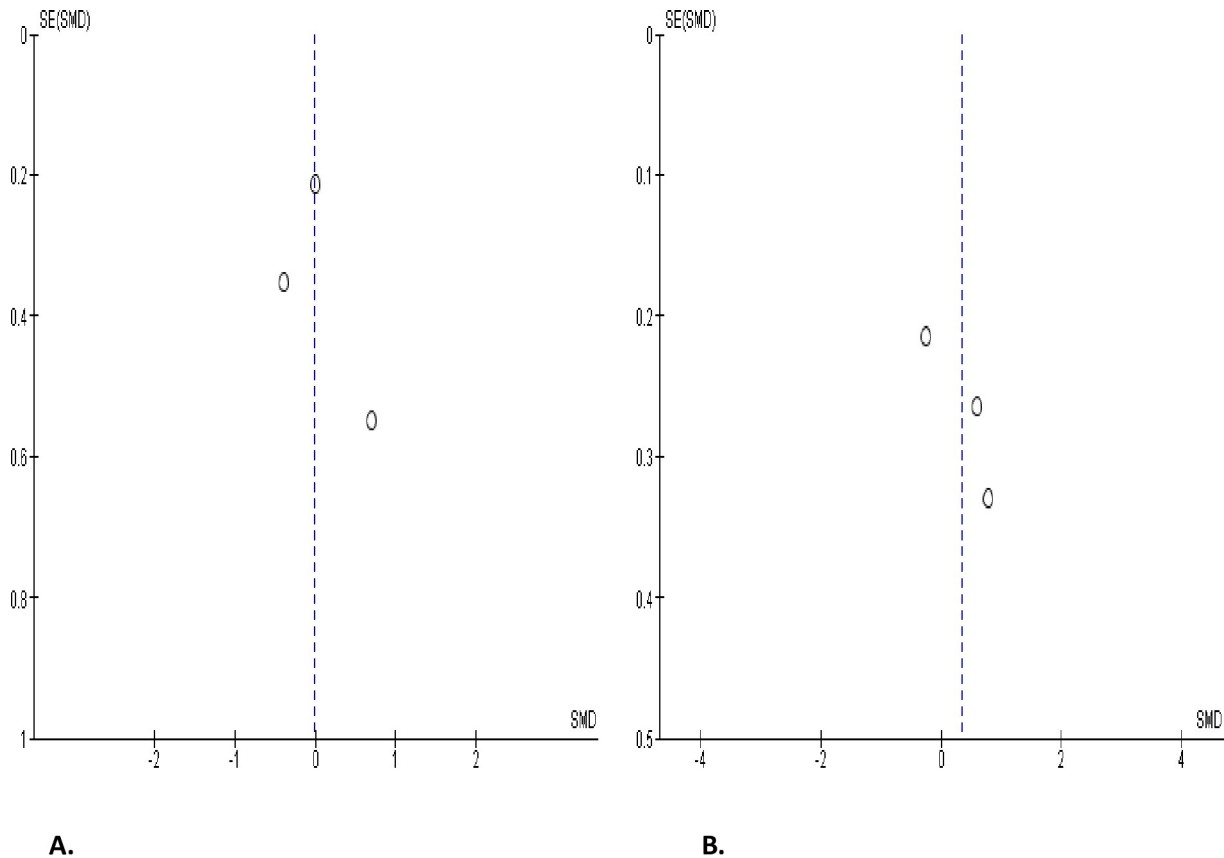

**A.**

**B.**

**Fig 17.** Funnel plot of seminal leukocyte count comparing between COVID-19 positive and COVID-19 negative patients (A) and before COVID-19 treatment and after COVID-19 treatment (B).

success rate of sperm retrieval in them. This implies that SARS-CoV-2 may lower the rate of spontaneous conception as well as reduce the success of ARTs. Since testosterone/LH is also a predictor of Leydig cell function [76,77], it is also credible to infer that SARS-CoV-2 impairs Leydig cell function. This may the reduced testosterone found in SARS-CoV-2 positive men.

It is imperative to note that the duration of the infection and time between infection and semen collection might have an effect on the study outcomes. Findings of Koç and Keseroğlu [48], and Temiz et al.[63] that performed semen analysis after 5 and 4 days of infection respectively showed insignificant changes for most of the sperm variables and testosterone level. It is also worth mentioning that most of the eligible studies were published between 2020 and 2022, indicating that they were likely before the introduction of COVID-19 vaccines and also before the infection by the most recent and less dangerous variants of COVID-19; hence, the impact of the virus may differ. It is likely that COVID-19 vaccination confers protection against sperm-endocrine aberrations induced by the novel virus. More so, the less virulent variants of COVID-19 may exert less adverse effect on the sperm-endocrine system than the virulent variant. Just like other systematic viral infections, SARS-CoV-2 impairs male fertility possibly by upregulating pro-inflammatory cytokines and promoting hyper-inflammation and oxidative stress or direct sperm-endocrine alterations [3]. The peculiarity of SARS-CoV-2 hinges around its novelty.

Despite the fascinating and convincing findings of this study, there are some limitations. First, the effect of SARS-CoV-2 on live-birth rate is not presented, which limits our conclusion

**A.**

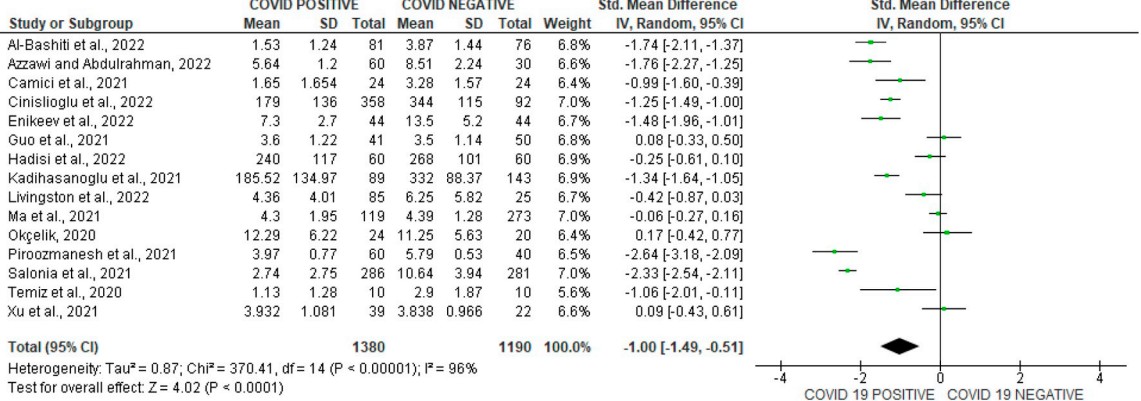

**B.**

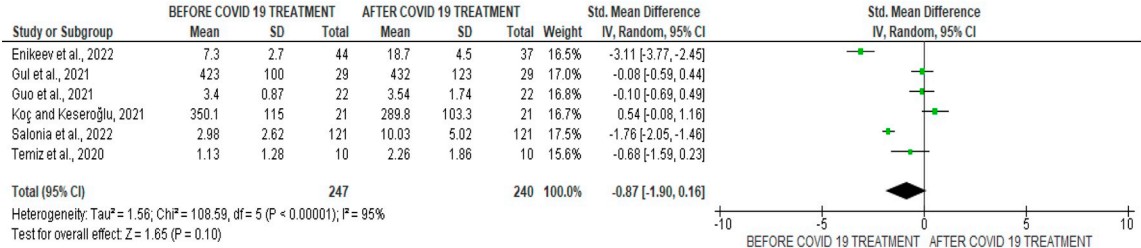

**Sensitivity analysis**

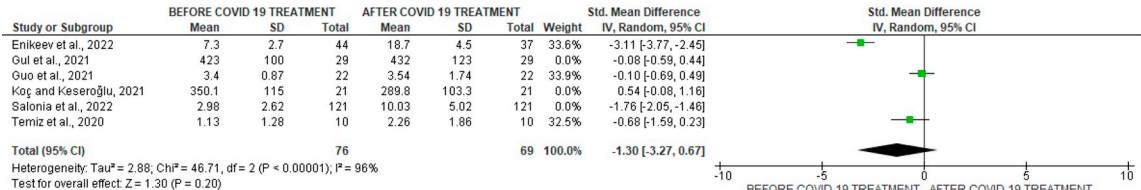

**C.**

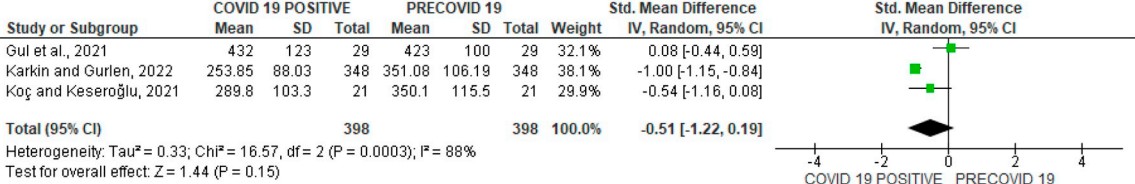

**Fig 18.** Forest plot of serum testosterone level comparing between COVID-19 positive and COVID-19 negative patients (A), before COVID-19 treatment and after COVID-19 treatment (B), and COVID-19 positive and preCOVID-19 period (C).

on the effect of the viral diseases on male fertility. Also, there were remarkable risk of publication bias in many of the studies. More so, the significant diversity in most of the studies is a major concern, although this was controlled by a sensitivity analysis. Lastly, studies exploring the actual mechanisms on SARS-CoV-2 on semen quality and male sex hormones are lacking and most studies were speculative. Nonetheless, the present meta-analysis provides an update and a robust data delineating the consequences of SARS-CoV-2 on conventional semen

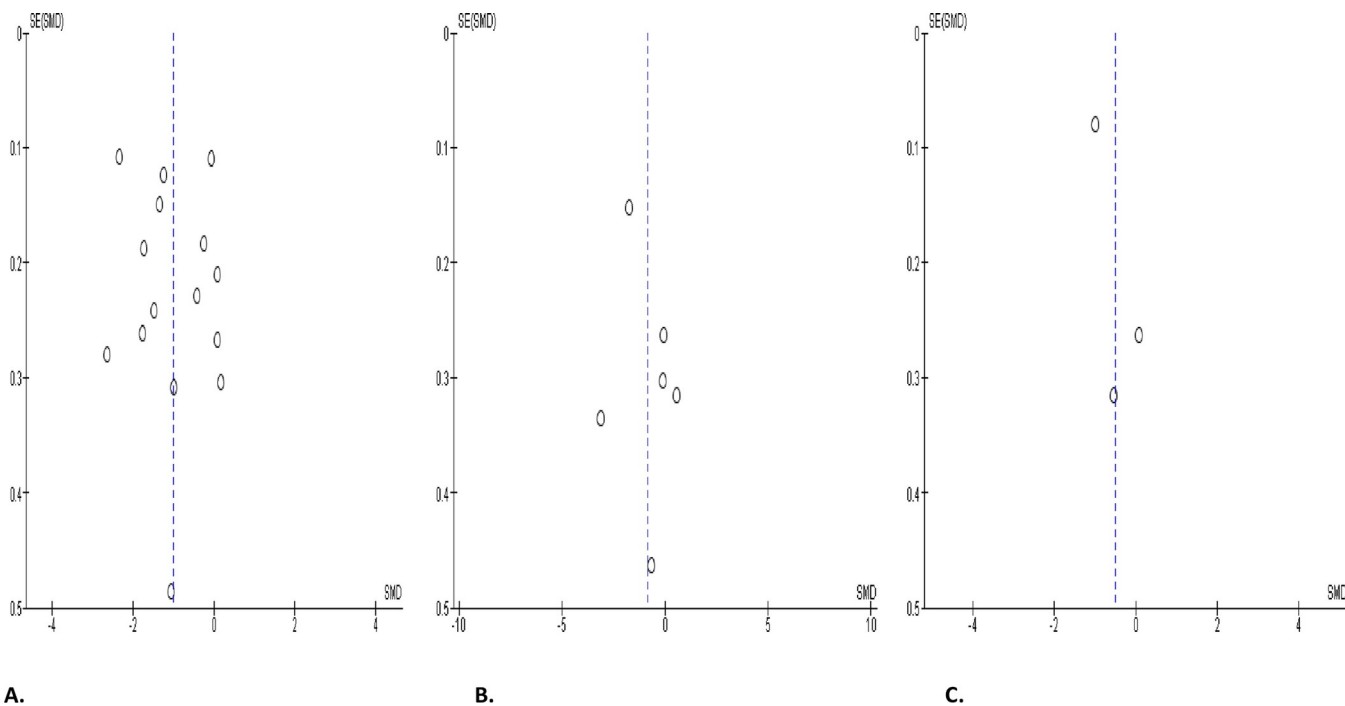

**Fig 19.** Funnel plot of serum testosterone level comparing between COVID-19 positive and COVID-19 negative patients (A), before COVID-19 treatment and after COVID-19 treatment (B), and COVID-19 positive and preCOVID-19 period (C).

parameters and male sex hormones. Detailed Strengths, Weaknesses, Opportunities, and Threats (SWOT) analysis of the current study is shown in Fig 29.

In conclusion, this study demonstrates that SARS-CoV-2 may diminish fertility in male by reducing semen quality viz. ejaculate volume, sperm count, concentration, viability, motility, and normal morphology through a hormone-dependent mechanism (reduction in testosterone level and increase in oestrogen and prolactin levels). It is also likely that the induction of oxidative stress and inflammatory injury play significant roles. More well-designed studies which accommodate larger sample size should be conducted to validate these findings, evaluate the long term effect of SARS-CoV-2 on sperm function and testosterone concentration, establish the associated mechanisms, and address the weaknesses highlighted are recommended.

**A.**

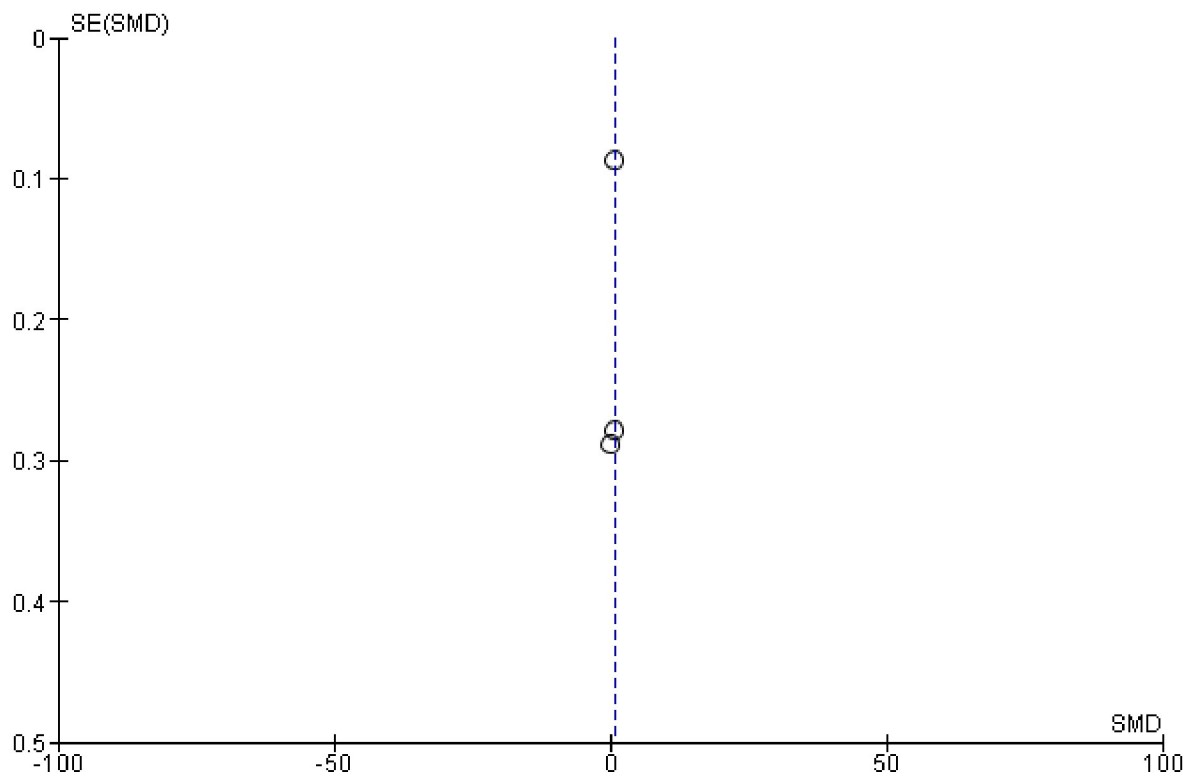

**B.**

**Fig 20.** Forest plot (A) and funnel plot (B) of serum oestrogen level comparing between COVID-19 positive and COVID-19 negative patients.

**A.**

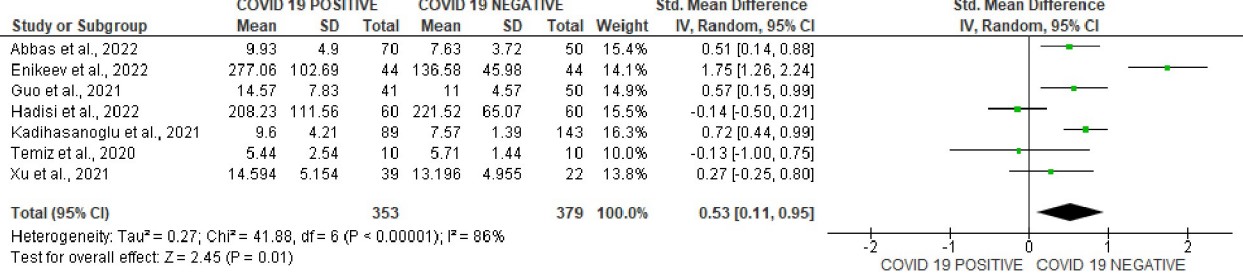

**B.**

**Fig 21.** Forest plot of serum prolactin level comparing between COVID-19 positive and COVID-19 negative patients (A) and before COVID-19 treatment and after COVID-19 treatment (B).

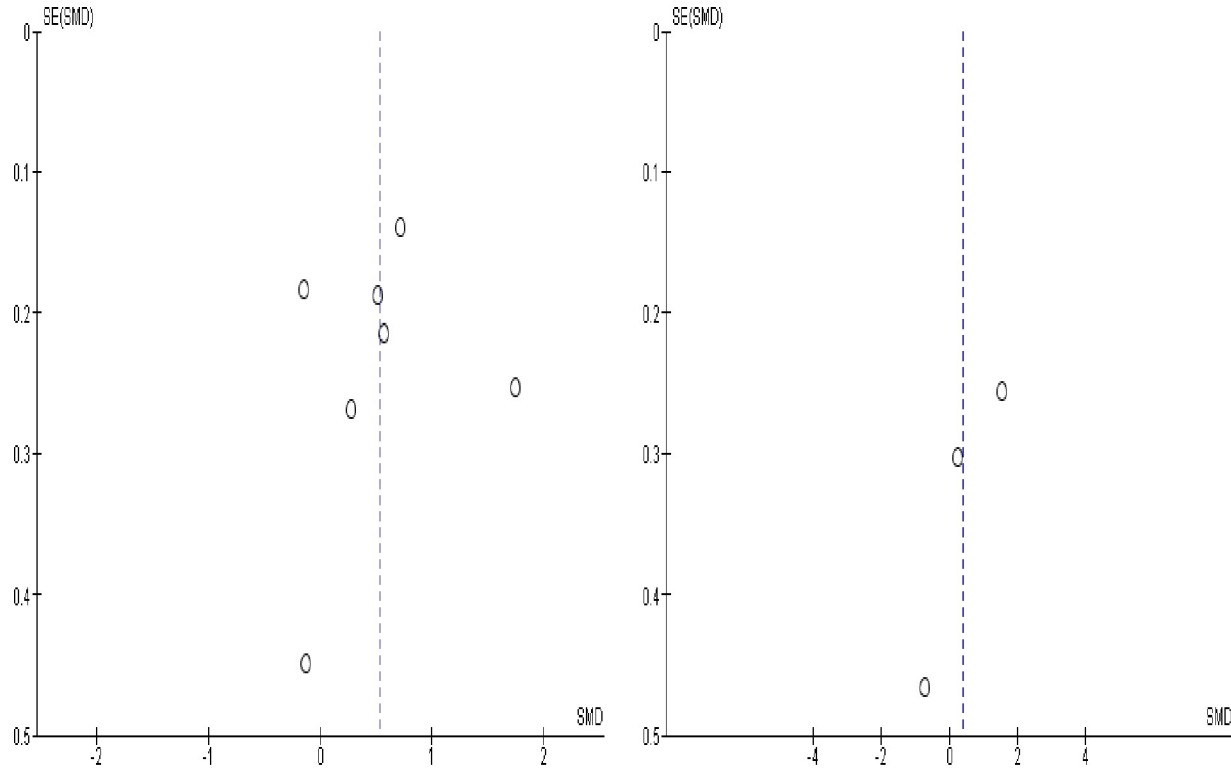

**A.**

**B.**

**Fig 22.** Funnel plot of serum prolactin level comparing between COVID-19 positive and COVID-19 negative patients (A) and before COVID-19 treatment and after COVID-19 treatment (B).

**A.**

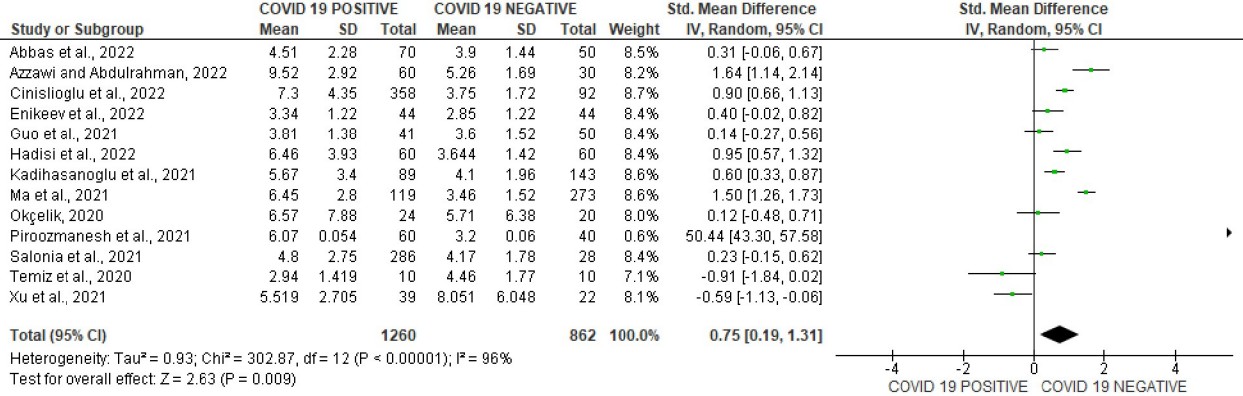

**Sensitivity analysis**

**B.**

**C.**

**Fig 23.** Forest plot of serum luteinizing hormone (LH) level comparing between COVID-19 positive and COVID-19 negative patients (A), before COVID-19 treatment and after COVID-19 treatment (B), and COVID-19 positive and preCOVID-19 period (C).

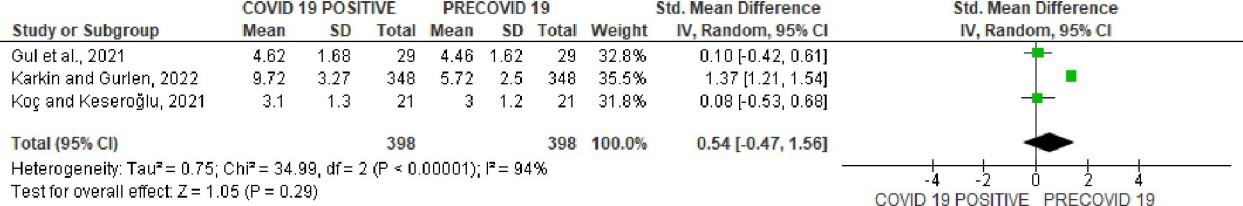

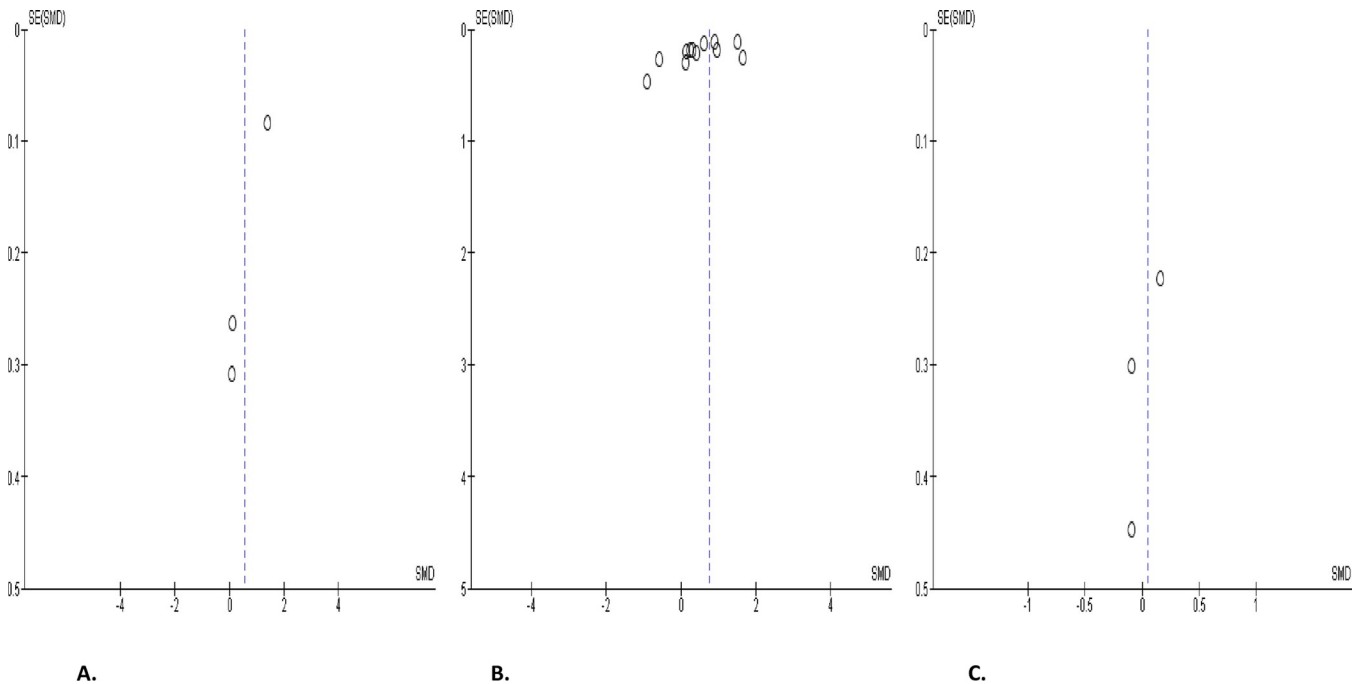

**Fig 24.** Funnel plot of serum luteinizing hormone (LH) level comparing between COVID-19 positive and COVID-19 negative patients (A), before COVID-19 treatment and after COVID-19 treatment (B), and COVID-19 positive and preCOVID-19 period (C).

**A.**

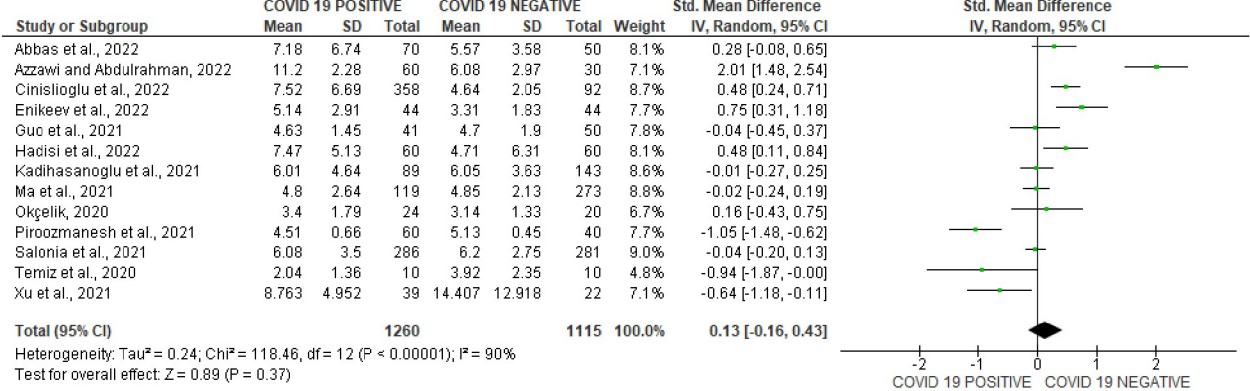

**Sensitivity analysis**

| Study or Subgroup | COVID 19 POSITIVE Mean | SD | Total | COVID 19 NEGATIVE Mean | SD | Total | Weight | Std. Mean Difference IV, Random, 95% CI |
|---|---|---|---|---|---|---|---|---|
| Abbas et al., 2022 | 7.18 | 6.74 | 70 | 5.57 | 3.58 | 50 | 0.0% | 0.28 [-0.08, 0.65] |
| Azzawi and Abdulrahman, 2022 | 11.2 | 2.28 | 60 | 6.08 | 2.97 | 30 | 8.7% | 2.01 [1.48, 2.54] |
| Cinislioglu et al., 2022 | 7.52 | 6.69 | 358 | 4.64 | 2.05 | 92 | 10.1% | 0.48 [0.24, 0.71] |
| Enikeev et al., 2022 | 5.14 | 2.91 | 44 | 3.31 | 1.83 | 44 | 9.3% | 0.75 [0.31, 1.18] |
| Guo et al., 2021 | 4.63 | 1.45 | 41 | 4.7 | 1.9 | 50 | 9.4% | -0.04 [-0.45, 0.37] |
| Hadisi et al., 2022 | 7.47 | 5.13 | 60 | 4.71 | 6.31 | 60 | 9.6% | 0.48 [0.11, 0.84] |
| Kadihasanoglu et al., 2021 | 6.01 | 4.64 | 89 | 6.05 | 3.63 | 143 | 10.0% | -0.01 [-0.27, 0.25] |
| Ma et al., 2021 | 4.8 | 2.64 | 119 | 4.85 | 2.13 | 273 | 10.2% | -0.02 [-0.24, 0.19] |
| Okçelik, 2020 | 3.4 | 1.79 | 24 | 3.14 | 1.33 | 20 | 8.4% | 0.16 [-0.43, 0.75] |
| Piroozmanesh et al., 2021 | 4.51 | 0.66 | 60 | 5.13 | 0.45 | 40 | 9.3% | -1.05 [-1.48, -0.62] |
| Salonia et al., 2021 | 6.08 | 3.5 | 286 | 6.2 | 2.75 | 281 | 0.0% | -0.04 [-0.20, 0.13] |
| Temiz et al., 2020 | 2.04 | 1.36 | 10 | 3.92 | 2.35 | 10 | 6.4% | -0.94 [-1.87, -0.00] |
| Xu et al., 2021 | 8.763 | 4.952 | 39 | 14.407 | 12.918 | 22 | 8.7% | -0.64 [-1.18, -0.11] |
| **Total (95% CI)** | | | 904 | | | 784 | 100.0% | 0.13 [-0.25, 0.51] |

Heterogeneity: Tau² = 0.35; Chi² = 113.71, df = 10 (P < 0.00001); I² = 91%
Test for overall effect: Z = 0.68 (P = 0.50)

**B.**

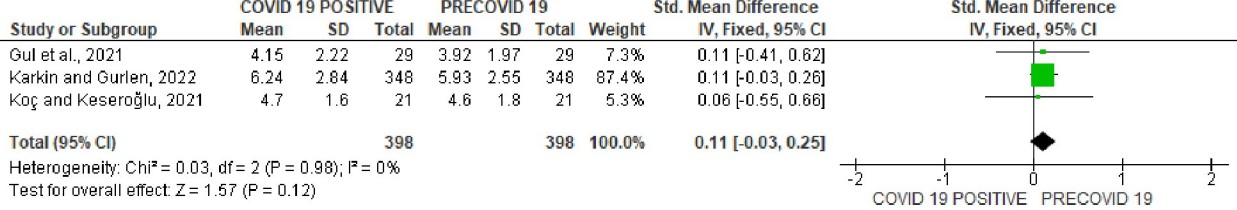

**C.**

| Study or Subgroup | COVID 19 POSITIVE Mean | SD | Total | PRECOVID 19 Mean | SD | Total | Weight | Std. Mean Difference IV, Fixed, 95% CI |
|---|---|---|---|---|---|---|---|---|
| Gul et al., 2021 | 4.15 | 2.22 | 29 | 3.92 | 1.97 | 29 | 7.3% | 0.11 [-0.41, 0.62] |
| Karkin and Gurlen, 2022 | 6.24 | 2.84 | 348 | 5.93 | 2.55 | 348 | 87.4% | 0.11 [-0.03, 0.26] |
| Koç and Keseroğlu, 2021 | 4.7 | 1.6 | 21 | 4.6 | 1.8 | 21 | 5.3% | 0.06 [-0.55, 0.66] |
| **Total (95% CI)** | | | 398 | | | 398 | 100.0% | 0.11 [-0.03, 0.25] |

Heterogeneity: Chi² = 0.03, df = 2 (P = 0.98); I² = 0%
Test for overall effect: Z = 1.57 (P = 0.12)

**Fig 25.** Forest plot of serum follicle-stimulating hormone (FSH) level comparing between COVID-19 positive and COVID-19 negative patients (A), before COVID-19 treatment and after COVID-19 treatment (B), and COVID-19 positive and preCOVID-19 period (C).

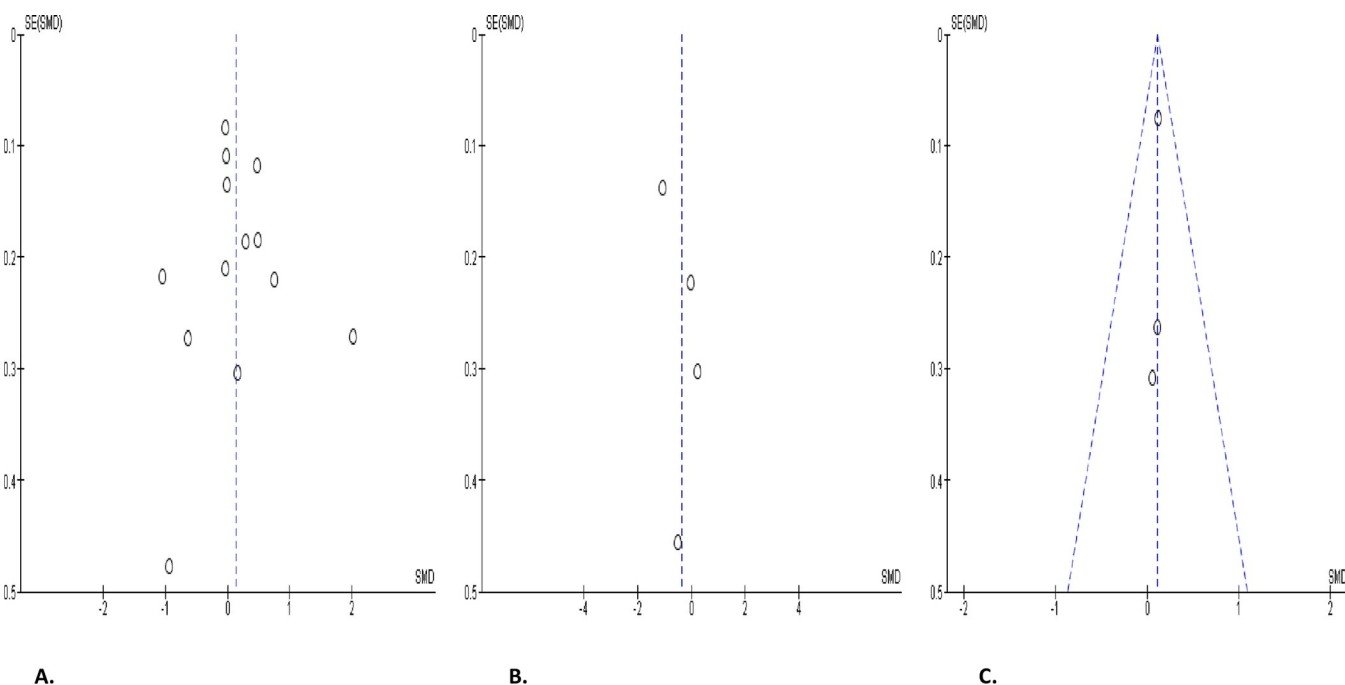

**Fig 26.** Funnel plot of serum follicle-stimulating hormone (FSH) level comparing between COVID-19 positive and COVID-19 negative patients (A), before COVID-19 treatment and after COVID-19 treatment (B), and COVID-19 positive and preCOVID-19 period (C).

**A.**

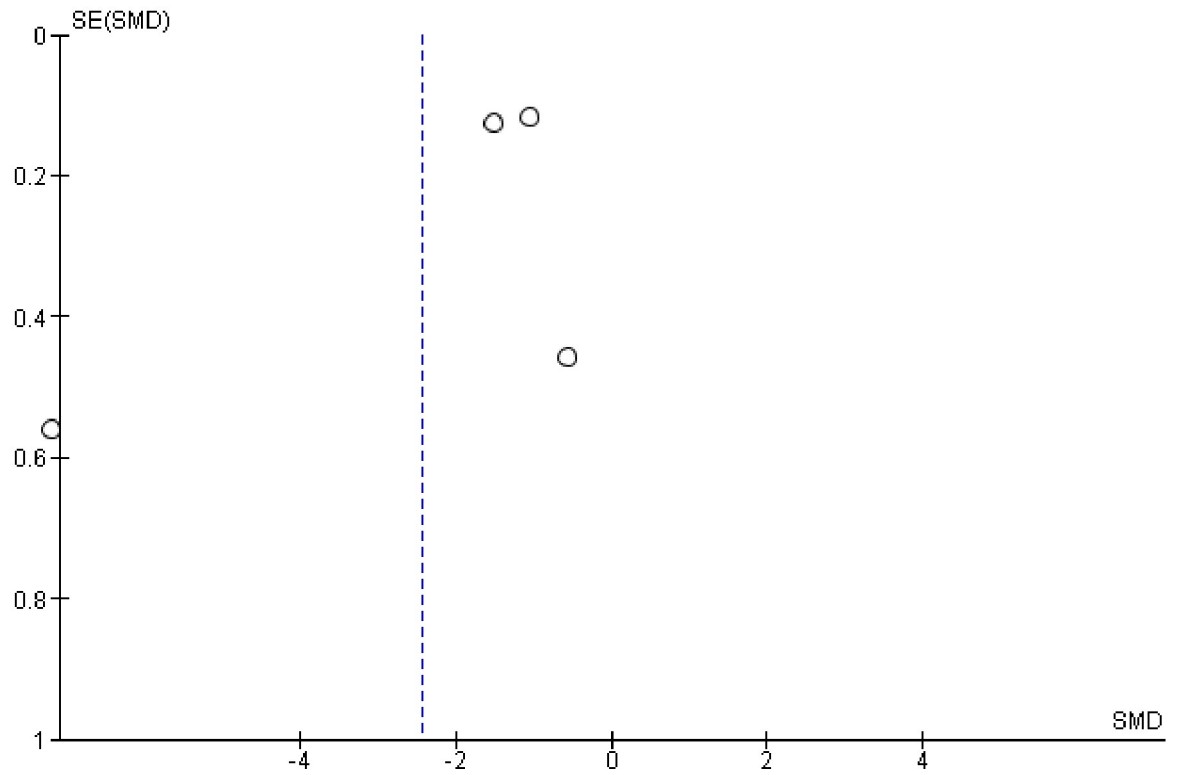

**B.**

**Fig 27.** Forest (A) and funnel (B) plots of serum testosterone/luteinizing hormone (T/LH) ratio comparing between COVID-19 positive and COVID-19 negative patients.

**A.**

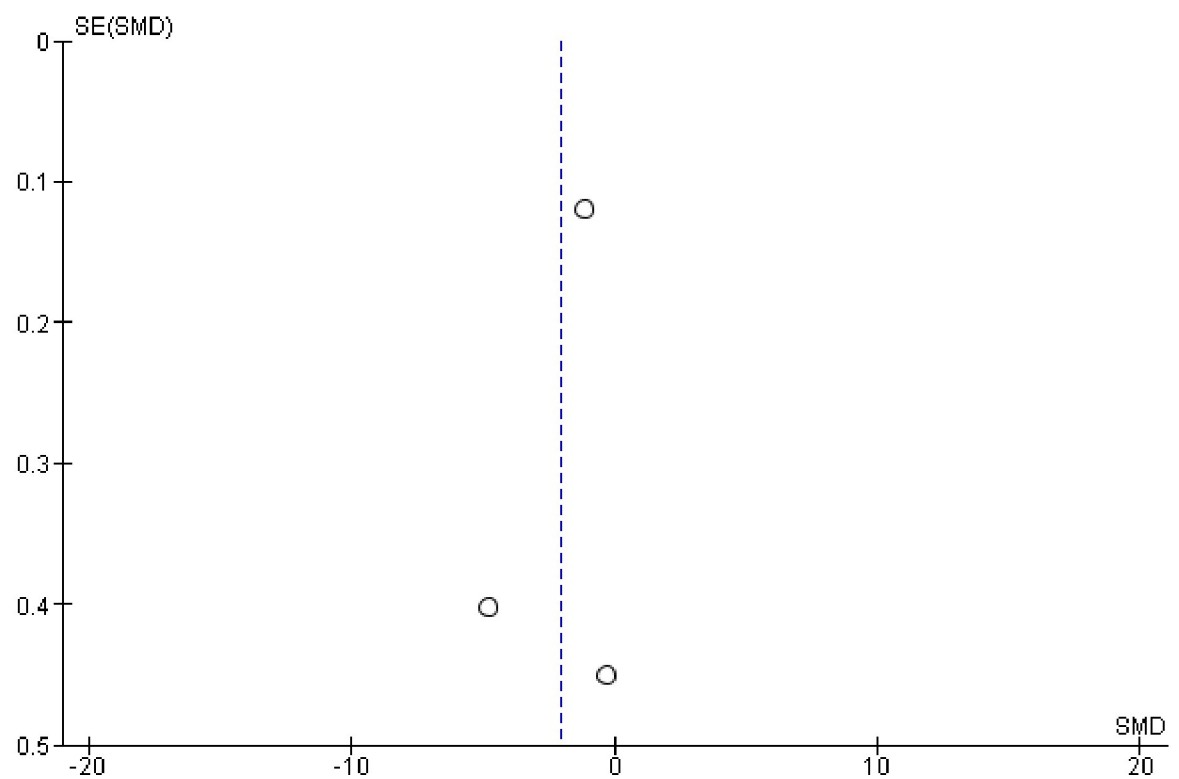

**B.**

**Fig 28.** Forest (A) and funnel (B) plots of serum follicle-stimulating hormone/luteinizing hormone (FSH/LH) ratio comparing between COVID-19 positive and COVID-19 negative patients.

**Fig 29. The Strengths, Weaknesses, Opportunities, and Threats (SWOT) analysis of the current study.**

## Supporting information

**S1 Checklist. PRISMA 2020 checklist.**
(DOCX)

**S1 Raw data.**
(ZIP)

## Author Contributions

**Conceptualization:** R. E. Akhigbe.

**Data curation:** R. E. Akhigbe.

**Formal analysis:** Akhigbe T. M., R. E. Akhigbe.

**Funding acquisition:** R. E. Akhigbe.

**Investigation:** Ashonibare V. J., Ashonibare P. J., Akhigbe T. M., R. E. Akhigbe.

**Methodology:** Akhigbe T. M., R. E. Akhigbe.

**Project administration:** Ashonibare V. J., Ashonibare P. J., Akhigbe T. M., R. E. Akhigbe.

**Resources:** R. E. Akhigbe.

**Supervision:** Akhigbe T. M., R. E. Akhigbe.

**Validation:** R. E. Akhigbe.

**Writing – original draft:** Ashonibare V. J., Ashonibare P. J., R. E. Akhigbe.

**Writing – review & editing:** Ashonibare V. J., Ashonibare P. J., Akhigbe T. M., R. E. Akhigbe.

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
