## [Decision Letter · Decision Letter 0]

17 Jun 2024

PONE-D-24-16835SARS-CoV-2 impairs male fertility by targeting semen quality and testosterone level: A Systematic Review and Meta-analysisPLOS ONE

Dear Dr. Akhigbe,

Thank you for submitting your manuscript to PLOS ONE. After careful consideration, we feel that it has merit but does not fully meet PLOS ONE’s publication criteria as it currently stands. Therefore, we invite you to submit a revised version of the manuscript that addresses the points raised during the review process.

I see high potential that this manuscript becomes an important reference. Both reviewers provide constructive suggestions for revisions.

We look forward to receiving your revised manuscript.

Kind regards,

Stefan Schlatt

Academic Editor

PLOS ONE

Journal Requirements:

https://www.frontiersin.org/journals/endocrinology/articles/10.3389/fendo.2023.1227836/full

In your revision ensure you cite all your sources (including your own works), and quote or rephrase any duplicated text outside the methods section. Further consideration is dependent on these concerns being addressed.

4. Please include your tables as part of your main manuscript and remove the individual files. Please note that supplementary tables (should remain/ be uploaded) as separate ""supporting information"" files

**Additional Editor Comments:**

This is an interesting paper describing the effects of a Covid infection on the male reproductive system. Both referees are very positive but have some suggestions for minor revsion.

Reviewers' comments:

Reviewer's Responses to Questions

**Comments to the Author**

1. Is the manuscript technically sound, and do the data support the conclusions?

Reviewer #1: Yes

Reviewer #2: Yes

2. Has the statistical analysis been performed appropriately and rigorously? 

Reviewer #1: Yes

Reviewer #2: Yes

3. Have the authors made all data underlying the findings in their manuscript fully available?

Reviewer #1: Yes

Reviewer #2: Yes

4. Is the manuscript presented in an intelligible fashion and written in standard English?

Reviewer #1: Yes

Reviewer #2: Yes

5. Review Comments to the Author

Reviewer #1: Dear Authors,

Congratulations for this interesting piece of research.

Some items worth considering:

1. It might be worth mentioning that duration of the infection and time between infection and semen collection might have an effect on the study outcomes. Based on current literature and on textbook knowledge, high fever might have a transient effect on semen analysis which might (or might not) be temporary.

2. Likewise, it might be worth mentioning if the studies were performed in early 2020, after introduction of COVID-19 vaccines, or following infection by the most recent and less dangerous variants of COVID-19.

3. Some of the studies were done before the release of the 2021 WHO manual. It might also be worth highlighting this difference, since it can affect the interpretation of the results, and maybe consider this when performing subgroup analyses.

Reviewer #2: The manuscript to be assessed is a systematic review and meta-analysis of the influence of SARS-CoV-2 infection on male fertility. There are already meta-analyses on this topic, but the authors explain very well that, in their view, there are methodological weaknesses in the already published meta-analyses that they have explicitly addressed in their work.

Overall, this is a methodologically very well conducted meta-analysis. The methodological steps and statistical analyses carried out are very well documented and openly communicated. The associated illustrations, especially the forest plots, are very detailed and comprehensive, which seems necessary due to the large number of parameters analysed.

Overall, in my view, this is a very mature manuscript for which I would like to congratulate the authors. The lack of originality in terms of existing publications is addressed by the authors and clearly explained what the methodological changes are here. As a suggestion for improvement, I would only add 2 aspects to the discussion.

1) I miss the discussion of the longitudinal aspect here, i.e. the duration of a potentially negative influence of SARS-CoV-2 infection on the reproductive functions of men, also with regard to the control intervals of the underlying studies.

2) in my view, it should also be discussed to what extent a SARS-CoV-2 infection represents a special feature compared to other systemically active viral infections, some of which are very severe (e.g. influenza), or whether this is not to be expected independently of the virus type.

6. PLOS authors have the option to publish the peer review history of their article (what does this mean?). If published, this will include your full peer review and any attached files.

Reviewer #1: No

Reviewer #2: No

---

## [Author Response · Author response to Decision Letter 0]

27 Jun 2024

Response: Thanks. This has been ensured.

https://www.frontiersin.org/journals/endocrinology/articles/10.3389/fendo.2023.1227836/full

In your revision ensure you cite all your sources (including your own works), and quote or rephrase any duplicated text outside the methods section. Further consideration is dependent on these concerns being addressed.

Response: Thanks. Overlapping phrases have been modified.

Response: Thanks. The Data Availability Statement is included as “All data are in the manuscript and/or supporting information files”.

4. Please include your tables as part of your main manuscript and remove the individual files. Please note that supplementary tables (should remain/ be uploaded) as separate ""supporting information"" files

Response: Thanks. The Tables have been added as part of the main manuscript.

Response: Thanks. 

Response: Thanks. The references have been checked for appropriateness.

Additional Editor Comments:

This is an interesting paper describing the effects of a Covid infection on the male reproductive system. Both referees are very positive but have some suggestions for minor revsion.

Response: Thanks. All concerns raised have been addressed.

Reviewers' comments:

Reviewer's Responses to Questions

Comments to the Author

1. Is the manuscript technically sound, and do the data support the conclusions?

Reviewer #1: Yes

Reviewer #2: Yes

2. Has the statistical analysis been performed appropriately and rigorously?

Reviewer #1: Yes

Reviewer #2: Yes

3. Have the authors made all data underlying the findings in their manuscript fully available?

Reviewer #1: Yes

Reviewer #2: Yes

4. Is the manuscript presented in an intelligible fashion and written in standard English?

Reviewer #1: Yes

Reviewer #2: Yes

5. Review Comments to the Author

Reviewer #1: Dear Authors,

Congratulations for this interesting piece of research.

Some items worth considering:

1. It might be worth mentioning that duration of the infection and time between infection and semen collection might have an effect on the study outcomes. Based on current literature and on textbook knowledge, high fever might have a transient effect on semen analysis which might (or might not) be temporary.

Response: Thanks. This has been included.

2. Likewise, it might be worth mentioning if the studies were performed in early 2020, after introduction of COVID-19 vaccines, or following infection by the most recent and less dangerous variants of COVID-19.

Response: Thanks. This has been included.

3. Some of the studies were done before the release of the 2021 WHO manual. It might also be worth highlighting this difference, since it can affect the interpretation of the results, and maybe consider this when performing subgroup analyses.

Response: Thanks. The WHO guideline would not affect the interpretation since the infective state was compared with a control (either COVID negative or PreCOVID or after treatment) and not just with the WHO standards.

Reviewer #2: The manuscript to be assessed is a systematic review and meta-analysis of the influence of SARS-CoV-2 infection on male fertility. There are already meta-analyses on this topic, but the authors explain very well that, in their view, there are methodological weaknesses in the already published meta-analyses that they have explicitly addressed in their work.

Overall, this is a methodologically very well conducted meta-analysis. The methodological steps and statistical analyses carried out are very well documented and openly communicated. The associated illustrations, especially the forest plots, are very detailed and comprehensive, which seems necessary due to the large number of parameters analysed.

Response: Thanks.

Overall, in my view, this is a very mature manuscript for which I would like to congratulate the authors. The lack of originality in terms of existing publications is addressed by the authors and clearly explained what the methodological changes are here. As a suggestion for improvement, I would only add 2 aspects to the discussion.

1) I miss the discussion of the longitudinal aspect here, i.e. the duration of a potentially negative influence of SARS-CoV-2 infection on the reproductive functions of men, also with regard to the control intervals of the underlying studies.

Response: Thanks. This has been included as suggested.

2) in my view, it should also be discussed to what extent a SARS-CoV-2 infection represents a special feature compared to other systemically active viral infections, some of which are very severe (e.g. influenza), or whether this is not to be expected independently of the virus type.

Response: Thanks. This has been included as suggested.

6. PLOS authors have the option to publish the peer review history of their article (what does this mean?). If published, this will include your full peer review and any attached files.

Do you want your identity to be public for this peer review? For information about this choice, including consent withdrawal, please see our Privacy Policy.

Reviewer #1: No

Reviewer #2: No

---

## [Decision Letter · Decision Letter 1]

4 Jul 2024

SARS-CoV-2 impairs male fertility by targeting semen quality and testosterone level: A Systematic Review and Meta-analysis

PONE-D-24-16835R1

Dear Dr. Akhigbe,

We’re pleased to inform you that your manuscript has been judged scientifically suitable for publication and will be formally accepted for publication once it meets all outstanding technical requirements.

Kind regards,

Stefan Schlatt

Academic Editor

PLOS ONE

Additional Editor Comments (optional):

Reviewers' comments:

Reviewer's Responses to Questions

**Comments to the Author**

1. If the authors have adequately addressed your comments raised in a previous round of review and you feel that this manuscript is now acceptable for publication, you may indicate that here to bypass the “Comments to the Author” section, enter your conflict of interest statement in the “Confidential to Editor” section, and submit your "Accept" recommendation.

Reviewer #1: All comments have been addressed

Reviewer #2: All comments have been addressed

2. Is the manuscript technically sound, and do the data support the conclusions?

Reviewer #1: Yes

Reviewer #2: Yes

3. Has the statistical analysis been performed appropriately and rigorously? 

Reviewer #1: Yes

Reviewer #2: Yes

4. Have the authors made all data underlying the findings in their manuscript fully available?

Reviewer #1: Yes

Reviewer #2: (No Response)

5. Is the manuscript presented in an intelligible fashion and written in standard English?

Reviewer #1: Yes

Reviewer #2: (No Response)

6. Review Comments to the Author

Reviewer #1: Dear Authors,

Thanks for addressing my previous concerns. I believe that this manuscript does not need any further edits.

Reviewer #2: Congratulations to this nice piece of science. After revision absolutely worthwhile for publication in my view

7. PLOS authors have the option to publish the peer review history of their article (what does this mean?). If published, this will include your full peer review and any attached files.

Reviewer #1: No

Reviewer #2: No

---

## [Editor Report · Acceptance letter]

17 Jul 2024

PONE-D-24-16835R1 

PLOS ONE

Dear Dr. Akhigbe, 

I'm pleased to inform you that your manuscript has been deemed suitable for publication in PLOS ONE. Congratulations! Your manuscript is now being handed over to our production team.

Kind regards, 

on behalf of

Dr. Stefan Schlatt 

Academic Editor

PLOS ONE